# Multiomics Investigation of Exhausted T Cells in Glioblastoma Tumor Microenvironment: *CCL5* as a Prognostic and Therapeutic Target

**DOI:** 10.3390/ijms26209920

**Published:** 2025-10-12

**Authors:** Ruihao Qin, Menglei Hua, Yaru Wang, Qi Zhang, Yong Cao, Yanyan Dai, Chenjing Ma, Xiaohan Zheng, Kaiyuan Ge, Huimin Zhang, Shi Li, Yan Liu, Lei Cao, Liuying Wang

**Affiliations:** 1Department of Biostatistics, School of Public Health, Harbin Medical University, Harbin 150086, China; 2022020127@hrbmu.edu.cn (R.Q.); 2022020217@hrbmu.edu.cn (M.H.); 2023020229@hrbmu.edu.cn (Y.W.); 2023020232@hrbmu.edu.cn (Q.Z.); 2022020110@hrbmu.edu.cn (Y.C.); 2022020216@hrbmu.edu.cn (Y.D.); 2022020124@hrbmu.edu.cn (C.M.); 2023020222@hrbmu.edu.cn (X.Z.); 2023020142@hrbmu.edu.cn (K.G.); 2023020145@hrbmu.edu.cn (H.Z.); 2Laboratory for the Study of Metastatic Microenvironments, Fred Hutchinson Cancer Research Center, Seattle, WA 98109, USA; sli7@fredhutch.org; 3School of Health Management, Harbin Medical University, Harbin 150086, China; wangliuying@hrbmu.edu.cn

**Keywords:** GBM, exhausted T cell, multiomics, drug sensitivity, *CCL5*

## Abstract

Glioblastoma multiforme (GBM) is a common malignancy with poor prognosis, and exhausted T (TEX) cells, a subset of T cells characterized by progressive loss of effector functions, play a critical role in its progression. This study aimed to investigate the impact of TEX-related genes on immune function, prognosis, and drug sensitivity in GBM through multiomics analysis. Initially, we identified a novel set of TEX-related genes specific to GBM and screened hub genes (*CCL5*, *IL18*, *CXCR6*, *FCER1G*, *TNFSF13B*) using conventional statistical methods combined with machine learning. A prognostic risk model was subsequently constructed based on TCGA data and validated in the CGGA cohort. Single-cell and pharmacogenomic analyses revealed significant differences in tumor microenvironment composition and drug sensitivity between risk groups. Notably, Palbociclib emerged as a potential therapeutic agent targeting the novel discovered biomarker *CCL5*. RT-qPCR results showed that T cells with low *CCL5* expression exhibited reduced expression of immune checkpoint-related genes (*PD1*, *TIM3*, *LAG3*) and increased expression of *CD28*, suggesting enhanced immune function. In conclusion, our findings highlight five hub genes as prognostic markers that could stratify GBM patients with different immune landscapes and levels of drug sensitivity. Furthermore, experimental results suggest that low *CCL5* expression could alleviate T cell exhaustion and represent a promising therapeutic target, offering new strategies for improving GBM prognosis.

## 1. Introduction

Exhausted T (TEX) cells represent a subset of T lymphocytes characterized by progressive loss of effector functions [1]. These cells demonstrate impaired self-renewal capacity, diminished immune function, metabolic dysregulation, and significantly elevated expression of inhibitory molecules, including *PD1*, *TIM3*, *BTLA*, *CTLA4*, and *LAG3* [1,2]. TEX cells pose a substantial challenge to current anti-cancer immunotherapies. However, blocking *PD1* can reinvigorate exhausted T cells, enhancing control over chronic infections and cancer. This underscores the potential of immune checkpoint inhibitors to reverse T cell depletion [3,4]. Meanwhile, CAR-T cells are also susceptible to exhaustion, which can compromise their therapeutic efficacy [5]. The widespread impact of T cell exhaustion on both natural and engineered immune responses has led to TEX cells garnering increasing attention in the study of disease-related immune microenvironments, with implications for improving current therapies and developing novel treatment strategies.

In the past, it was believed that the brain was a closed organ with immune privilege [6]. However, recent research has fundamentally changed the view of brain immunity, revealing its close interaction with the immune system and the presence of a unique immune microenvironment [6,7,8,9]. GBM has a dismal 5-year survival rate and accounts for approximately 54.4% of all malignant gliomas [10,11,12]. As the most aggressive central nervous system tumor, the persistent antigen exposure from GBM may induce severe T cell exhaustion [9,13,14]. This exhaustion could undermine immune function and affect the survival of GBM patients. Meanwhile, Temozolomide (TMZ) remains the standard treatment for GBM, but development of resistance in over 50% of patients necessitates the exploration of novel biomarkers and therapeutic mechanisms [15,16,17]. Given the potential reversibility of T cell exhaustion, targeting TEX-related genes could not only identify prognostic indicators and form the basis for risk stratification and immune landscape characterization but also provide new immune checkpoints and therapeutic targets. However, there has been no integrative study analyzing the status and biomarkers of T cell exhaustion in GBM and its impact on the immune microenvironment and drug sensitivity.

In this study, we aimed to investigate the impact of T cell exhaustion on immune ability, prognosis, and drug resistance in GBM patients through multiomics analysis. First, we employed the Mendelian randomization (MR) method to establish the potential causal relationship between glioblastoma and exhausted T cells. Then, we performed differential gene expression analysis between glioblastoma and normal brain tissue, and between TEX and Teff cells. We identified the intersecting genes and employed protein–protein interaction (PPI) networks, survival models, and proteomics to further refine key genes (Figure 1A). Based on hub genes, we constructed a risk model using a TCGA database for GBM prognosis and validated it using an independent cohort from CGGA (Figure 1B). Through single-cell omics and pharmacogenomics, we observed significant differences in cellular components, immune microenvironment, signaling pathways, and drug sensitivity between the high- and low-risk groups. Furthermore, our analysis suggested that Palbociclib may have potential therapeutic value in GBM (Figure 1C). Pseudotime analysis indicated a potential regulatory role for *CCL5* in T cell exhaustion in GBM. To validate this, we performed in vitro experiments, including *CCL5* knockout in T cells. Finally, RT-qPCR results confirmed our hypothesis regarding *CCL5*’s role in T cell exhaustion in GBM (Figure 1D).

## 2. Results

### 2.1. Mendelian Randomization Analysis

Mendelian randomization, utilizing meta-analysis of published genome-wide association studies (GWAS) on glioma, identified a potential causal relationship between glioma and an increased proportion of exhausted non-naive T cells within the CD8^+^ T cell population (*p* = 0.0233) (Appendix A). The *p*-values for both the horizontal pleiotropy test and Cochran Q test were greater than 0.05, indicating the impact of horizontal pleiotropy and heterogeneity on the results was negligible (Appendix A). Then, GWAS summary statistics data of a GICC study and UCSF/Mayo study about GBM were utilized to enhance the credibility of this conclusion. Both results demonstrated that gliomas impact the proportion of CD28^−^ CD8^+^ T cells (*p* of GICC study = 0.0329, *p* of UCSF/Mayo study = 0.0449). These additional analyses supported the significance of the causal effect (Table 1). Concurrently, neither the Cochran Q test nor the horizontal pleiotropy test revealed any significant differences (Appendix A). The smaller β-value indicated that GBM exerts a slow effect on exhausted T cells, consistent with the understanding that T cell exhaustion requires prolonged antigen stimulation. These findings provide evidence suggesting that GBM may contribute to T cell depletion within the brain microenvironment.

### 2.2. Construction of TEX-Related Gene Set in GBM

Using the GEO dataset (GSE234100), which includes TEX cells and their corresponding effector cells, we identified 1286 genes associated with TEX (Figure 2A,B). The Wilcoxon rank sum test was used to analyze the differential expression in normal brain tissues and GBM, revealing 2834 significantly differentially expressed genes. The intersection of these two sets resulted in 361 genes potentially influencing T cell exhaustion in glioblastoma, which were considered as the GBM gene set related to TEX (Figure 2C).

### 2.3. PPI Network Construction and Hub Gene Selection

To further elucidate the correlation and potential interactions among genes in the TEX-related gene set of GBM, we constructed a PPI network using the STRING database (Figure 3A,B). The resulting PPI network, comprising 273 nodes and 3424 edges, was then analyzed using Cytoscape. This analysis revealed that Cluster1 and Cluster2 had significantly higher MCODE scores (32.686 and 18.267, respectively) compared to other clusters (Figure 2D), and these two clusters comprised a total of 67 hub genes (Appendix A). The hub genes include well-known factors such as *TNF*, *PDCD1*, *CXCR6*, and *CXCL10*, as well as other important genes like *CDC20*, *CDC45*, *DTL*, and *EXO1* [18].

According to the information of GO enrichment analysis and KEGG pathway analysis performed in hub genes, we found that Cluster1 was tightly associated with cell cycle and Cluster2 had an effect on immunity [18,19,20] (Figure 3A,B). The GeneMANIA results indicated that the genes in Cluster1 were not only related to the cell cycle and DNA replication but also participated in the transduction of the p53 signaling pathway. *CCNB1*, *CDC25C*, *PCNA*, *FOX1*, and *EXO1* were all involved in the p53 signaling pathway, a pivotal regulator in anti-tumor immunity that modulates the expression of numerous target genes. This pathway could significantly impact immune cell functions and the immune response within the tumor microenvironment [21]. The depletion of DNA repair genes such as *EXO1* had been confirmed to result in the buildup of DNA damage and genomic instability [22,23], which may be related to the difficulty in reversing T cell exhaustion. *CCNB1* and *CDC25C* were the most important hub genes because they were involved in multiple functional pathways within Cluster1. They might lead to T cell exhaustion by reducing the cell cycle signatures and p53 signaling transduction [21]. Compared to Cluster1, the predominant functions of Cluster2 were primarily related to immune cell migration and chemotaxis, regulation of immune cell activation, cell surface receptor expression, cytokine receptor binding, and response to biostimulation. Evidently, the interactions of Cluster2 were more directly related to T cell exhaustion. The interactions of Cluster2 might trigger T cells to migrate towards the tumor region and upregulate the expression of inhibitory surface receptors. Subsequently, through cytokine-driven stimulation, these processes could impair the activation capacity of T cells, prompting their transition into exhausted T cells, ultimately fostering immune evasion in glioblastoma. We found that *CCL5* and were prominently involved in numerous functions, suggesting their potential significance as key genes within Cluster2. Additionally, the interaction of *PDCD1* on T cells binding with PD-L1 on tumor cells enables the evasion of tumor cell recognition by T cells [24], Tumor necrosis factor (*TNF*) inhibits tumor cell growth [25], and *CD28* mediates co-stimulation of T cells, enhancing their survival, proliferation, and cytokine production [26]. They were all identified as the first neighbors of *CCL5* [27]. *CCL5* also exhibited close neighboring relationships with *CD68* and *KLRK1* [27,28], with *CD68* serving as the most reliable marker for macrophages and *KLRK1* activating the innate immune response of natural killer (NK) cells [29,30]. The associations suggested potential interactions between exhausted T cells and both NK cells and macrophages in the tumor microenvironment.

### 2.4. Construction of the GBM Prognostic Risk Model

To explore the associations between hub genes and patient prognosis, we developed a prognostic risk model. The GBM cohort from TCGA contained 161 patients with clinically relevant information. To minimize the influence of confounding factors, we employed multivariable Cox regression analysis on all hub genes, identifying 30 significant genes with *p*-values < 0.05 (Table 2, Figure 4D). We then applied the Random Survival Forest (RSF) model, XGBoost, and LightGBM for survival analysis to further refine prognostic risk factors among the hub genes. For the XGBoost (Figure 4E) and LightGBM (Figure 4F) models, we selected the top half of the variable importance (VIMP) values for further analysis. Additionally, the RSF model revealed that 28 genes had a positive prognostic effect (Figure 4A). By intersecting results from all four models, we identified a set of five key genes (*CXCR6*, *IL18*, *FCER1G*, *CCL5*, *TNFSF13B*) consistently associated with GBM prognosis.

These key genes were utilized to construct a risk model. We calculated the TEX risk score with the following formula: risk score = *CXCR6* × (0.232258970) + *IL18* × (−0.026403607) + *FCER1G* × (0.005410601) + *CCL5* × (−0.023922295) + *TNFSF13B* × (0.005089899). Patients were stratified into two risk groups based on the median risk score (median 0.0781). Kaplan–Meier survival curves demonstrated that the OS of the low-risk group was significantly better than that of the high-risk group (*p* = 0.00018) (Figure 5A). To assess the sensitivity and specificity of the prognostic model, time-dependent ROC curves were generated, and the area under the ROC curve (AUC) was calculated: 0.629 at 400 days, 0.640 at 800 days, 0.721 at 1200 days, and 0.8159 at 1600 days (Figure 5E). Further validation was performed using an independent dataset from the Chinese Glioma Genome Atlas (CGGA). The verification dataset yielded similar results (*p* = 0.0014) (Figure 5D), although the AUC was lower at 400 days (Figure 5F). We then integrated risk group information with other relevant clinical factors and applied a Cox regression model to examine their significance. In the TCGA Set, we analyzed six clinical factors (Age, Ethnicity, Sex, Race, Pharmaceutical Therapy, Radiation Therapy) and found significantly decreased mortality rates in patients classified as low-risk (*p* = 0.0015) or those who received radiation therapy (*p* < 0.001) (Figure 5G). In the CGGA Set, our Cox regression model incorporated seven clinical factors (Sex, Age, Radiation Therapy, Pharmaceutical Therapy, IDH mutation status, 1p19q codeletion status, MGMTp methylation status), demonstrating that patients in the low-risk group had significantly lower mortality risk (*p* = 0.0135) (Figure 5H). The decision curve demonstrated that two survival models yielded a higher net benefit than both the “treat none” and “treat all” strategies in most time points of the TCGA Set, especially in the early stages (Appendix A). However, as disease progressed, both models showed clear advantages over the “treat all” strategy only at higher threshold probability (Appendix A). Further, the prognostic model including risk group and clinical information performed better than the prognostic model that only included the risk group across a wider range of risk thresholds (Appendix A). The DCA results for the CGGA Set were generally consistent with those from the TCGA Set (Appendix A).

### 2.5. Proteomic Results

We further extended our investigation to the proteomic level to gain deeper insights into the specificity of key genes in GBM. Our proteomic analysis, based on data from the Clinical Proteomic Tumor Analysis Consortium (CPTAC), focused on *CCL5*, *IL18*, and *FCER1G*. The results revealed significantly higher protein abundance of these markers in cancer compared to normal tissues (both *p* < 0.0001) (Appendix A). Additionally, *FCER1G* displayed a strong correlation with *IL18* (Appendix A), suggesting potential functional relationships. Pan-cancer analysis showed that *IL18* and *CCL5* proteins were overexpressed in GBM compared with other tumor types (Figure 5I,J). *FCER1G* also showed increased expression (Appendix A). Notably, while *CCL5* expression was downregulated in most tumor types, it showed the highest expression levels in GBM, indicating its unique role in GBM progression.

### 2.6. The TME and Functional Features of Two Risk Groups

The proteomic results prompted us to investigate potential differences in tumor characteristics and immune responses between two risk groups. Utilizing single-cell omics data, we inferred the composition of malignant cells and various non-malignant cell types within the tumor microenvironment (TME) (Figure 6A,B). The results revealed significant differences in malignant cells, myeloid cells, and pericytes; in particular, tumor infiltration was significantly higher in the high-risk group (*p* = 0.0183) (Figure 6F). In contrast, the composition of T cells, endothelial cells, and oligodendrocytes was similar between the two groups. By correlating key genes with *PDCD1* and *TNF* as references, we found that *IL18* and *TNFSF13B* might contribute to an increase in the total number of T cells (both *p* < 0.01) (Figure 6E). Interestingly, while *CCL5*, *CXCR6*, and *FCER1G* did not affect the total T cell count, CIBERSORT analysis revealed that they exhibited significant negative correlations with activated NK cells (−0.43, −0.45, and −0.37, respectively) (both *p* < 0.001) and resting CD4^+^ memory T cells (−0.33, −0.26, and −0.35, respectively) (both *p* < 0.001) and positive correlation with CD8^+^ T cells (0.36, 0.43, and 0.23, respectively) (*p* of *CCL5* and *CXCR6* < 0.001, *p* of *FCER1G* < 0.01) and activated CD4^+^ memory T cells (0.34, 0.32, and 0.26, respectively) (both *p* < 0.001) (Figure 7B–D and Appendix A). This suggested that their expression might be associated with the stimulation and differentiation of T cells and NK cells. Further analysis using multiple algorithms (TIMER, EPIC, MCPcounter, and xCell) revealed distinct TME profiles between the two risk groups. The high-risk group generally had fewer CD8^+^ T cells across most algorithms, except for CIBERSORT (Figure 7E–J). ssGSEA analysis indicated that the high-risk group had more exhausted T cells (*p* < 0.01) (Figure 8A) and NK cells (*p* < 0.01) (Appendix A), higher expression of immune checkpoints (*p* < 0.01) (Figure 8B), and more active cancer-immunity cycles (Figure 8I and Appendix A). Additionally, pathway analysis revealed that processes such as Pan-F-TBRS (*p* < 0.01), EMT1 (*p* < 0.05), EMT2 (*p* < 0.01), and EMT3 (*p* < 0.05) were notably enhanced in the high-risk group, indicating increased epithelial–mesenchymal transition (EMT) (Appendix A).

Further investigation into immune function pathways revealed differences in TCR signaling pathways between the two risk groups (*p* < 0.01) (Figure 8C). Additionally, patients in the high-risk group exhibited higher secretion of IFNγ (*p* < 0.01) (Figure 8D) and leukocyte antigens [31] (*p* < 0.01) (Figure 8G,H), suggesting the five key genes might influence the immune function through multiple pathways, contributing to T cell exhaustion. In contrast, TGFβ and its receptor scores remained similar between the two groups (both *p* > 0.05) (Figure 8E,F), indicating that these genes are unlikely to act through this pathway.

Combining findings of BayesPrism, we found that although the total number of brain T cells in the high-risk group did not differ significantly from that in the low-risk group, the high-risk group had a higher proportion of exhausted T cells. Additionally, we believed that differential expression of *CCL5* might not affect the total number of T cells in the brain, but affect the immune microenvironment by impairing T cell function, leading to poorer prognosis.

### 2.7. Drug Sensitivity

We evaluated the drug sensitivity of patients in different risk groups using pharmacogenomic data from the CTRP and PRISM databases. Vandetanib showed significant differences between the two risk groups across both datasets (*p* < 0.05 in CTRP and *p* < 0.001 in PRISM). In contrast, Temozolomide only exhibited differences in the CTRP dataset (*p* < 0.001). Notably, procarbazine in the PCV chemotherapy (*p* < 0.01) and everolimus (*p* < 0.05) demonstrated higher sensitivity in the high-risk group. This pattern was also observed with several adjuvant and second-line drugs commonly used in GBM treatment (Figure 9A,B and Appendix A).

Subsequently, we identified five known drugs (Camptothecin, MK-1775, Nutlin-3a(-), Palbociclib, and VE821) that may target the key proteins (*CCL5*, *IL18*, *CXCR6*, *FCER1G*, *TNFSF13B*) (Figure 9C). Among these, Camptothecin was found to inhibit DNA topoisomerase I (Topo I), while MK-1775 targets cyclin dependent kinase 1 (*CDK1*). We then analyzed pharmacokinetic characteristics of these candidate drugs using ADMETlab2.0, which indicated that Palbociclib, the first *CDK4/6* inhibitor for breast cancer treatment with efficient blood–brain barrier (BBB) penetration, held promise for potentially assisting in GBM treatment through the identified target genes (Figure 9D). To further validate the feasibility of Palbociclib treatment, we performed molecular docking with the key proteins. The docking models revealed particularly strong intermolecular interactions between Palbociclib and both *CCL5* and *IL18*, with estimated free energy of binding values of −7.06 kcal/mol and −8.89 kcal/mol, respectively (Figure 9E,F and Appendix A). Proteomics results showed that *CCL5* and *IL18* proteins exhibited the most significant increase in expression in the brain compared to other cancer types. Candidate drugs targeting these two proteins, such as Palbociclib, might offer more potential therapeutic benefits for GBM treatment.

### 2.8. Pseudotime Analysis Results

To further investigate the regulatory role of key genes in T cell exhaustion, we performed pseudotime analysis using the GSE210534 dataset, which captured four distinct T cell states in chronological order: resting T cells (Trest), effector T cells (TEFF), tumor-infiltrating T cells (Ttumor), and exhausted T cells (TEX). In the pseudotime model, we selected three individuals and incorporated their bulk gene expression data across these different cellular states. According to this model, we observed that the expression levels of *CCL5* were consistently higher in the TEX phase compared to the TEFF phase. Furthermore, except for *IL18*, other key genes showed a progressive increase in expression over time, starting from the Ttumor phase. Among these, *CCL5* showed the most pronounced upregulation (Figure 9G). This temporal expression pattern of *CCL5*, particularly its abnormal increase, was consistent with the progression of T cell exhaustion. The exceptionally high expression of *CCL5* observed in proteomics analysis, coupled with its significant position in the PPI network and pseudotime model, led us to hypothesize a potential regulatory role for *CCL5* in T cell exhaustion.

### 2.9. Verification by RT-qPCR

To confirm that *CCL5* was involved in the regulation of T cell exhaustion, we conducted in vitro experiments. The cultured T cells were divided into five groups. Two groups served as controls: one as standard control and the other as negative control (Figure 10B). The remaining three groups were subjected to different siRNAs targeting the *CCL5* gene for knockdown experiments. Among these three T cell groups transfected with different siRNAs (siRNA-323, siRNA-234, and siRNA-416), the siRNA-416 group had the lowest *CCL5* expression (Figure 10A,C). Consequently, this group was selected for further analysis using the CD3/CD28 magnetic bead stimulation assay. After 15 days of magnetic bead exposure, RT-qPCR analysis revealed significant changes in gene expression (Appendix A). Compared to T cells in the two control groups, the expression levels of *PD1*, *TIM3*, and *LAG3* were markedly decreased (both *p* < 0.001) in the *CCL5* knockdown group (Figure 10D,E). Meanwhile, the expression level of *CD28* was significantly higher in the *CCL5* knockdown group compared to the other two groups (*p* < 0.001) (Figure 10F). *CD28* is crucial for T cell activation; its increased expression implies a potential enhancement of T cell effector capabilities. The detailed gene primer sets are provided in Appendix A.

KEGG database and several studies suggested that *CCL5* may regulate the NFκB pathway [32,33,34,35]. Previous research had demonstrated that NFκB activation promotes T cell exhaustion [36,37]. Moreover, a bidirectional regulatory relationship existed between *CCL5* and IFNγ, where IFNγ enhanced *CCL5* secretion while simultaneously contributing to T cell exhaustion [31,38]. Based on these findings, we hypothesized that *CCL5* may modulate T cell exhaustion through the NFκB signaling pathway (Figure 10H). To test this hypothesis, we stratified TCGA patients into high and low *CCL5* expression groups based on mean expression. Our analysis revealed significantly enhanced NFκB pathway activation in patients with high *CCL5* expression (*p* < 0.01) (Figure 10G). Therefore, we believed that in glioblastoma, the overexpression of *CCL5* enhances the NKκB and IFNγ signaling pathway of tumor-infiltrating T cells, leading to their exhaustion. Alterations in the cell surface receptors of T cells result in immune evasion, ultimately promoting tumor progression (Figure 10I).

These results provided evidence that *CCL5* knockdown slows the process of T cell exhaustion. Moreover, T cells appeared to maintain a high immune capacity at low *CCL5* expression. This observation supported the regulatory role of *CCL5* in T cell exhaustion. Additionally, our speculation on the primary regulatory pathway of *CCL5* offered insights into potential therapeutic strategies for reinvigorating exhausted T cells in the tumor microenvironment of GBM.

## 3. Discussion

GBM remains one of the most aggressive cancers, largely due to its limited effective treatment options and the immune evasion mechanisms employed by the tumor [39,40,41]. Despite advances in understanding the molecular mechanisms of GBM, drug resistance still exists, and there is an urgent need to explore more effective targets [42,43,44]. T cell exhaustion is characterized by the progressive loss of effector functions, including cytokine production and cytotoxic activity, in tumor-infiltrating T cells. Studies on other tumors have found that T cell exhaustion promotes tumor immune escape [45], and therapies targeting this mechanism have shown initial success in some tumors [46]. However, due to the existence of the brain immune privilege theory [6], research on the immune microenvironment of GBM started late. Consequently, there is a lack of research on identifying genes that could alleviate T cell exhaustion in the brain to treat GBM.

Therefore, this study comprehensively investigated key genes associated with T cell exhaustion and their impact on prognosis, immune capacity, and drug sensitivity in GBM through multiomics analysis. Utilizing these key genes, we constructed a reliable prognostic model and stratified GBM patients into high- and low-risk groups of T cell exhaustion. Subsequent analyses revealed significant differences between risk groups in terms of their immune landscape and drug sensitivity, providing valuable insights for guiding potential treatment strategies in GBM. Furthermore, we proposed candidate drugs targeting these biomarkers and obtained the pharmacokinetics characteristics to assess their ability. Notably, our in silico and cell experiments provided novel evidence that *CCL5*, one of our identified key genes, bound to the proposed candidate drug and was involved in the regulation of T cell exhaustion. These findings might inform the development of improved therapeutic strategies for GBM.

Our Mendelian randomization analysis suggested that GBM might induce severe T cell exhaustion. As MR revealed only potential causal relationships, we utilized several datasets. Through genomic analysis and a PPI network, we obtained a hub gene set for TEX in GBM. We further identified five key genes (*CCL5*, *IL18*, *CXCR6*, *FCER1G*, *TNFSF13B*) by using both machine learning and traditional statistical methods. *CCL5* is known to be involved in immune cell recruitment and exhaustion, and may play dual roles in tumor progression [47,48,49]. *IL18* is a cytokine that is involved in the activation of immune responses [50]. The expression of *FCER1G*, *CXCR6*, and *TNFSF13B* also demonstrated important roles in immune modulation. Specifically, *CXCR6* and *FCER1G* are involved in the trafficking and activation of immune cells [51,52], while *TNFSF13B* has been implicated in the regulation of immune responses [53]. Accordingly, we constructed a reliable prognostic risk model and stratified GBM patients into high- and low-risk groups based on the risk score from this model. DCA results indicated that the survival model included both risk groups, and other clinical information was more suitable as a prognostic model, especially in the early stages.

Our proteomic analysis confirmed the overexpression of *CCL5*, *IL18*, and *FCER1G* in GBM compared to normal tissues. Notably, *CCL5* showed significantly higher levels in brain tumors, suggesting a unique role in GBM. *CCL5* was also found to be strongly correlated with *IL18*, but our study did not explore the interactions between them in depth.

We subsequently investigated the impact of TEX on the immune microenvironment and the potential contributions of key genes. Results of BayesPrism showed *IL18* and *TNFSF13B* might contribute to an increase in the total number of T cells. Analysis of the tumor immune microenvironment revealed that the high-risk group exhibited a significant increase in the number of exhausted T cells, which correlated with the poor prognosis of GBM patients. Based on these findings, we believed that differential expression of *CCL5* might not affect the total number of T cells in the brain, but affect the immune microenvironment by impairing T cell function, leading to poorer prognosis. Our analysis also found that the expression of key genes was associated with other immune cells. However, we did not further investigate whether the key genes were involved in cell communications between T cells and other immune cells.

We also performed drug sensitivity analysis using pharmacogenomic data from the CTRP and PRISM databases to assess the therapeutic potential of the two risk groups. We found that the high-risk group showed increased sensitivity to Temozolomide and PCV treatments. We further identified Palbociclib as a potential therapeutic agent. The pharmacokinetic characteristics demonstrated that Palbociclib exhibited efficient blood–brain barrier penetration (BBB penetration of 0.22). Molecular docking results indicated that Palbociclib had strong binding affinity to *CCL5* and *IL18*, with estimated free energy of binding values of −7.06 kcal/mol and −8.89 kcal/mol, respectively. Additionally, previous studies suggested the use of Palbociclib in brain metastases from breast cancer [54,55], leading us to hypothesize that its combination with TMZ would improve treatment outcomes. These findings supported its feasibility as an adjuvant therapeutic drug. However, the present results were based solely on pharmacogenomic data and docking simulations, without further experimental validation of the potential of the candidate drug in combination therapy.

Previous studies presented conflicting views on the role of *CCL5* in T cell function. While some research suggested that high *CCL5* expression was associated with T cell exhaustion in other cancers [56,57,58], other studies reported that high *CCL5* expression in tumors correlates with increased T cell activity [38,47]. Our pseudotime analysis provided novel insights. During the TEFF phase, *CCL5* expression remained at a relatively low level, but it gradually increased during exhaustion. This indicated that *CCL5* might play a dual role in the regulation of T cell immunity. But pseudotime analysis did not represent an actual biological time sequence; the roles of other key genes might have been neglected.

RT-qPCR results showed that T cells with low *CCL5* expression exhibited reduced expression of immune checkpoint-related genes (*PD1*, *TIM3*, *LAG3*) and increased expression of *CD28*, suggesting enhanced immune function. Integrating these findings, we believed that *CCL5* could recruit T cells and enhance immune capacity at low expression levels [47,59], while at higher levels, it might promote T cell exhaustion [58]. This dual role of *CCL5* offered a novel understanding of its impact on T cell behavior and the immune microenvironment in GBM. We also observed that high *CCL5* expression was associated with increased activity of the NFκB signaling pathway. This process likely predominated in T cell exhaustion in GBM [37], but still lacked further experimental validation.

Although our study primarily relied on bioinformatics analysis, it made several significant contributions to the field. Firstly, we identified a novel TEX-related gene set specific to GBM and explored its hub genes and associated transcription factors. This discovery had potential implications for reversing T cell exhaustion and enhancing the efficacy of ICB and CAR-T therapies. However, constraints in sequencing depth and sample size might lead to bias or censoring of some TEX cell genes, potentially overlooking their biological roles. Furthermore, we constructed a risk model based on TEX-related genes in GBM and stratified patients into two risk groups. Meanwhile, pan-cancer proteomic analysis showed that *CCL5* protein was upregulated in GBM, exhibiting the greatest difference compared to other cancers. Multiomics analysis revealed significant differences between the two risk groups in both the immune microenvironment and drug sensitivity. Then, we identified the potential candidate drug Palbociclib, which could guide clinical treatment strategies. Experimental results further demonstrated that *CCL5* was involved in the regulation of T cell exhaustion; specifically, reducing *CCL5* expression could delay this process. Given the above results, we suggested that *CCL5* could be a promising therapeutic target in GBM treatment. Intriguingly, KEGG pathway results indicated that *CCL5* might induce T cell exhaustion through the NFκB pathway rather than the TGFβ pathway [37,60]. We also observed that high *CCL5* expression was associated with increased activity of the NFκB signaling pathway, with this process likely predominating in T cell exhaustion in GBM. This view provided a novel research direction for reversing T cell exhaustion in GBM.

Nevertheless, our findings regarding the regulatory effects and reversal mechanisms relied mainly on RT-qPCR analysis, lacking additional experiments to further validate these conclusions, such as in-depth mechanistic or animal experiments. Also, our study focused on GBM, without considering other glioma types or T cell subsets. Future studies will further investigate the role of *CCL5* in cell communication between T cell subsets or with other immune cells, conduct additional experiments, such as mechanism experiments and in vivo models, and consider other glioma types.

## 4. Materials and Methods

### 4.1. Dataset Collection

We utilized six previously reported GWAS SNP datasets related to glioma from the NHGRI-EBI GWAS catalog [61], along with GWAS summary statistics data for GBM from the GICC study and UCSF/Mayo study [62]. Additionally, public GWAS data related to TEX were obtained from the MR Base database (Appendix A) [63]. Then, we collected RNA sequencing data of human T cell subsets, including hypofunctional exhausted cells (TEX) and fully activated effector cells (TEFF), from the Gene Expression Omnibus (GEO) database. Specifically, we utilized dataset GSE234100, which comprises four TEX samples and four effector T cell samples [19]. The TCGA-GBM cohort containing RNA sequencing and corresponding clinical information of 167 patients was downloaded from the TCGA website (https://portal.gdc.cancer.gov/, accessed on 15 November 2023). As an external independent cohort for validation, 325 patients from the Chinese Glioma Genome Atlas (CGGA) database (http://www.cgga.org.cn/, accessed on 20 November 2023) were included in the study [64,65,66]. Normal brain tissue data were sourced from paracancerous tissue samples in TCGA, with additional normal brain tissue data obtained from The Genotype-Tissue Expression (GTEx) project [67]. Lastly, to provide a single-cell perspective, we employed a dataset (GSE103224) of 23,793 cells with 3′ end-enriched tag clusters from 8 GBM patients as a reference [68] (Table 3).

### 4.2. Mendelian Randomization

To investigate the potential causal effect of GBM on the exhaustion process of T cells, we employed a two-sample Mendelian randomization (MR) method. *CD28* is a co-stimulatory molecule expressed on the surface of T lymphocytes, playing a crucial role in T cell activation, and the loss of *CD28* expression has been confirmed to be associated with severe T cell exhaustion [26,69]. T cells expressing *CD45RA* are considered naive and lack effector functions, whereas CD45RA-negative T cells represent mature effector populations [70,71]. Therefore, we used CD28^−^ CD8^+^ T cells and CD45RA^−^ CD28^−^ CD8^+^ T cells from the GWAS catalog to represent TEX cells as the outcome. This approach enabled us to capture both the general exhausted phenotype (CD28^−^) and the subset that becomes exhausted after exerting immune functions (CD45RA^−^ CD28^−^).

The genome-wide significant SNP threshold was set at *p* < 5.0 × 10^−6^, and the F-statistic was used to measure the strength of the association between IVs and the exposure. The *F*-statistic was calculated using the formula $F=\frac{R^{2}\times (N-2)}{1-R^{2}}$, where $R^{2}=2\times MAF\times (1-MAF)\times {\left(\frac{\beta }{sd}\right)}^{2}$, and SNPs with an *F*-statistic greater than 10 were retained to avoid the influence of weak instrument bias [72,73]. We first utilized meta-analyzed GWAS data on glioma as the exposure and employed a two-sample MR using the MR Base platform (app.mrbase.org). The multiplicative random-effects inverse variance weighted (IVW) method served as the primary approach for Mendelian randomization (MR) analysis, with statistical significance defined as *p* < 0.05. To ensure the validity and robustness of the results, sensitivity analyses were implemented using three additional MR methods, namely weighted median, simple mode, and weighted mode. The MR-Egger method offered an intercept test to detect horizontal pleiotropy, with a significance threshold of *p* < 0.05 indicating the presence of horizontal pleiotropy. Additionally, the Cochran Q test was utilized to examine heterogeneity, with a significance threshold of *p* < 0.05 indicating significant heterogeneity. The analysis parameters included a minimum linkage disequilibrium (LD) R-squared value of 0.8 and a minor allele frequency (*MAF*) threshold of 0.3 for aligning palindromic SNPs [74]. To further validate our findings, we conducted additional analyses using GWAS summary statistics from the GICC study and the UCSF/Mayo study as the exposure, focusing specifically on GBM patients. For these two validation datasets, we utilized the “TwoSampleMR” package (version 0.5.11) and the “MRInstruments” package (version 0.3.2) within R software (version 4.3.1).

### 4.3. Identification of TEX-Related DEGs

The dataset GSE234100 was used to screen for differentially expressed genes (DEGs) between TEX and TEFF cells. The exactTest function (from the “edgeR” package) was used for comparison between groups, which was suitable for the small sample dataset. *p*-values were adjusted for multiple comparisons using the False Discovery Rate (FDR) method. We set the biological coefficient of variation (BCV) at 0.4 (typical value of human data), and applied the following criteria: |Log FC| > 1.8, FDR < 0.005, and |Log CPM| > 0.5 [75,76,77]. Log fold change meant the logarithm of the fold change in gene expression between TEX and TEFF cells, with a higher value indicating a greater relative difference in gene expression between the two samples. Log CPM could be understood as the logarithm of counts per million, representing the expression level of a gene in every million cells, which was used to filter out lowly expressed genes [75]. Due to the small sample size, we lowered the FDR threshold to obtain more meaningful genes. The “edgeR” package (version 4.0.2) was used [76].

### 4.4. Identification of Common DEGs in GBM

To identify DEGs in GBM, we performed differential expression analysis for mRNA, lncRNA, and miRNA between GBM and normal brain tissue. The Wilcoxon rank-sum test was used to minimize the risk of exaggerated false positives, which were often associated with commonly used differential expression methods [78]. *p*-values were adjusted for multiple comparisons using the FDR to control the false positive rate. Differentially expressed genes were defined by the following criteria: |Log FC| > 2 and FDR  <  0.05. To obtain a TEX-related gene set in GBM, we intersected the differential gene sets from the TEX/TEFF analysis with those from the GBM tissue analysis.

### 4.5. PPI Network, GO, and KEGG Analysis Based on Common DEGs

Protein–protein Interaction (PPI) networks were established by the Search Tool for the Retrieval of Interacting Genes Database (STRING, http://string-db.org/, accessed on 1 December 2023), and module analyses were performed using Cytoscape software (version 3.10.1). The parameters of the MCODE plugin were set to the default parameters. Subnetworks with an MCODE score greater than 10.0 were considered networks containing hub genes.

### 4.6. Survival Analysis

Prior to model construction, we performed necessary data preprocessing. For patient mortality outcomes, we utilized available death dates, setting the survival status to 1 when present. In cases where the death date was missing or unclear, we imputed using the date of last follow-up.

Multivariate Cox proportional hazard regression analysis (R package “survival”, version 3.8.3), Random Survival Forest model [79] (R package “randomForestSRC”, version 3.3.1), XGBoost with AFT model [80] (R package “xgboost”, version 1.7.8.1) and LightGBM model [81] (R package “lightgbm”, version 4.5.0) were used to screen key genes in the TCGA cohort. We screened genes with a *p*-value < 0.05 in the Cox regression and genes with variable importance in projection (VIMP) > 0 in Random Survival Forest. We also selected genes at the top half of VIMP in the XGBoost and LightGBM models. VIMP measured the importance of each feature on the model’s predictive outcome, with a higher value indicating the gene had a greater impact on the prognosis [79,80]. To establish the risk model, we identified the intersection of the above genes. The risk score formula was as follows: $\sum_{i}^{N}\left(coefficient\left(i\right)\times expression\left(i\right)\right)$, where i represented the genes selected as covariates. We stratified GBM patients in both TCGA sets and CGGA validation sets into high- and low-risk groups based on the median risk score. Subsequently, we conducted survival analysis and compared the survival differences between the two risk groups using the log-rank test. The receiver operating characteristic (ROC) curves were generated by the “survminer” (version 0.5.0), and “timeROC” (version 0.4) R packages to evaluate the accuracy of the survival model. The decision curve analysis (DCA) was performed by the “ggDCA” R package (version 1.2) to evaluate the clinical utility of the survival model.

### 4.7. Proteomics Analysis

Proteomics analysis was conducted to further validate the specificity of key genes in glioblastoma. We used proteomic data from the National Cancer Institute’s Clinical Proteomic Tumor Analysis Consortium (CPTAC) [82]. The unpaired *t*-test (*p*-value less than 0.05 was considered statistically significant) was utilized to identify the differences in key protein abundance between GBM and normal brain tissues. Correlation analysis between key proteins was also performed. Moreover, we extended our analysis to compare core protein abundance across multiple cancer types to identify glioblastoma-specific therapeutic targets.

### 4.8. Cell Proportion Reconstruction and Function Analysis

The Bayesian model, using the GSE103224 dataset as prior information, was utilized to infer the cell type fractions of patients in the TCGA cohort. The results were then used to explore correlations with key genes and to identify differences between the high- and low-risk groups. Pearson’s correlation coefficient was calculated to assess the correlations, with higher values indicating stronger relationships. Additionally, the *p*-value was computed using Student’s *t*-test, with a value less than 0.05 considered statistically significant. The Bayesian model and related single-cell reference preprocessing were computationally estimated using the “BayesPrism” (version 2.1.1) R package [83].

Tumor immune microenvironments (TMEs) were estimated using the CIBERSORT [84,85], xCell [86], EPIC [87,88], MCPcounter [89], and TIMER [90,91]. Functional gene sets included immune function, anti-cancer immunity status, and features of immunodepletion. Activity scores were calculated using single sample gene set enrichment analysis (ssGSEA) from the “GSVA” package (v1.50.0). Differences in these characteristics between the two risk groups were compared using the Wilcoxon rank-sum test, and a *p*-value < 0.05 was considered statistically significant.

### 4.9. Candidate Drug Analysis

To investigate differential treatment responses between the two risk groups in GBM, we conducted a comprehensive drug sensitivity analysis using pharmacogenomics data from two established databases, PRISM and CTRP. The “pRRophetic” (version 0.5) package, which implemented a ridge regression model, was utilized to evaluate drug responses for GBM patients. This approach yielded estimated AUC values for each compound across all patients. K-nearest neighbor (k-NN) imputation was applied to impute the missing AUC values, and lower AUC values indicated increased sensitivity to treatment. We utilized the Wilcoxon rank-sum test to reveal the differences in drug sensitivity between the two risk groups, and a *p*-value < 0.05 indicated significant difference. Additionally, Pearson’s correlation was calculated between AUC and risk scores.

To identify potential adjuvant drugs targeting key proteins, we used drug sensitivity data from the GDSC database. Focusing on GBM-related cell lines and H9 T lymphocytes, we integrated these data with our model’s risk coefficients to reconstruct drug response profiles. The R package “pRRophetic” (version 0.5) was used to calculate AUC values and *p*-values for the drug response of each patient. We retained drugs that demonstrated lower AUC values than Temozolomide in both GBM and T cell lines. Potential drug candidates were identified based on two criteria: a negative correlation between AUC and risk score, and a *p*-value < 0.05 [92]. Subsequently, we obtained the pharmacokinetics and safety profiles of the drug candidates from ADMETlab2.0 (https://admetmesh.scbdd.com, accessed on 11 March 2024). AutoDock (version 4.2.6) and PyMOL were utilized to further analyze the interactions between the identified drugs and key proteins.

### 4.10. Pseudotime Analysis

To further elucidate the regulatory role of key genes, we employed pseudotime analysis using the GSE210534 dataset. In the pseudotime model, each T cell state represented a sample point. The model performed PCA analysis on the sample points, calculated the Euclidean distance between them to generate the timeline, and scaled the timeline to 0~10 to explain temporal relationships of key genes [93,94]. The pseudotime model was constructed by R package “bulkPseudotime” (version 0.0.1).

### 4.11. Transfection and Quantitative Real-Time Polymerase Chain Reaction

We considered human T lymphocyte lineage as the experimental subject and performed cell transfection with synthesized siRNAs (siRNA-323, siRNA-234, siRNA-416) to inhibit expression of *CCL5* in T cells. The group with the highest transfection efficiency was selected for further analysis. T cells were stimulated with CD3/CD28 magnetic beads (Gibco Dynabead™ Human T-Activator CD3/CD28, 11161D) for 15 days [95,96], and RNA was reverse-transcribed into cDNA using the RevertAid First Strand cDNA Synthesis Kit (K1622). qRT-PCR analysis was performed using the 2× SYBR Green qPCR Master Mix (B21203) on an ABI7500 Real-time System. More details are provided in Appendix A. We performed the Wilcoxon test to compare the relative expression values between groups and corrected the *p*-values using the FDR method.

## 5. Conclusions

This study identified five key TEX-related genes—*CCL5*, *IL18*, *CXCR6*, *FCER1G*, and *TNFSF13B*. These genes could predict survival in glioblastoma patients and stratified them into two risk groups. Multi-omics analysis revealed significant differences between the two risk groups in terms of tumor microenvironment and drug sensitivity. Notably, these findings suggested that these novel prognostic markers could serve as targets for immune checkpoint inhibitors or CAR-T immunotherapy to reverse T cell exhaustion. Thus, this research proposed potential drugs targeting these biomarkers. *CCL5* was particularly important, and experiments showed that reducing its expression could slow down T cell exhaustion. Targeting *CCL5* and its associated pathways might represent a promising strategy to enhance immune responses and improve patient outcomes in GBM.

## Figures and Tables

**Figure 1 ijms-26-09920-f001:**
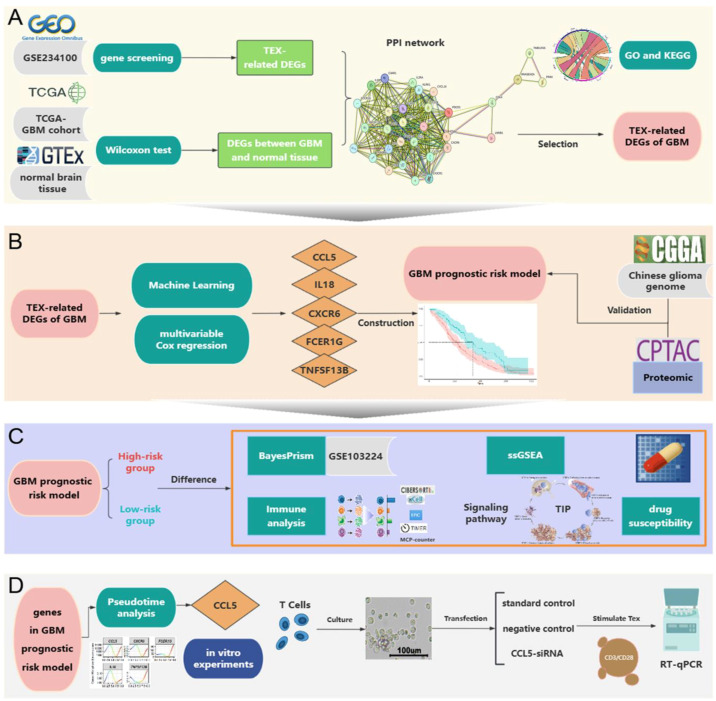
General overview of this research. (**A**) Utilizing GEO, TCGA, and GTEx to identify common DEGs in GBM and selecting hub genes via PPI network. (**B**) Establishing a GBM prognosis risk model based on the identified DEGs and validating this model using CGGA and proteomics data. (**C**) Using the prognosis model to stratify GBM patients into high- and low-risk groups, followed by comprehensive analysis of inter-group disparities across multiple aspects and identification of potential therapeutic agents targeting the model’s constituent genes. (**D**) Investigating how genes within the prognostic model regulate T cell exhaustion through pseudotime analysis. Findings were validated with in vitro experiments.

**Figure 2 ijms-26-09920-f002:**
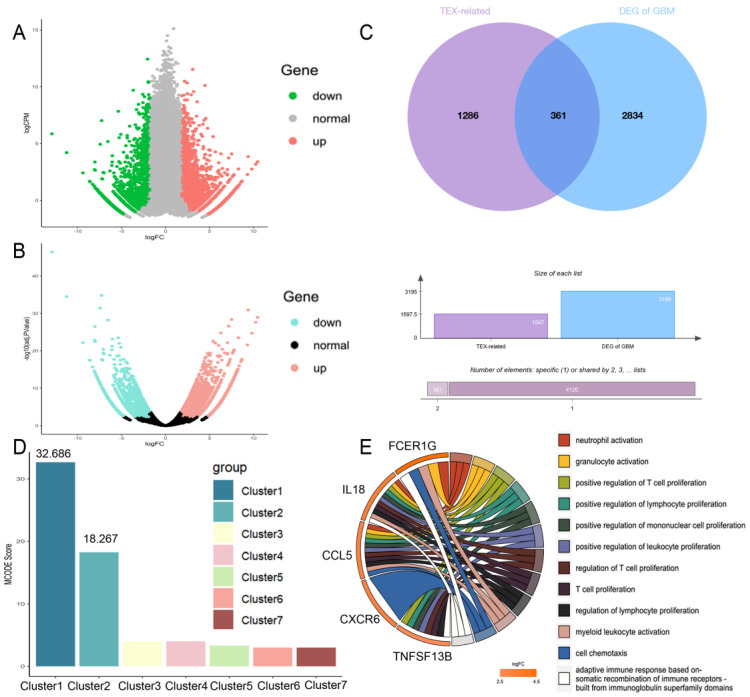
Identification of TEX-related gene list based on GBM patients. (**A**) Bland–Altman plot of the DEGs between TEX cells and effective T cells from GEO datasets. Green represents downregulated DEGs, and red represents upregulated DEGs. (**B**) Volcano plot of the DEGs between TEX cells and effective T cells from GEO datasets. Skyblue represents significant downregulated DEGs, red represents significant upregulated DEGs, and black represents no difference. (**C**) Venn diagram demonstrates the identification of TEX-related gene list based on GBM patients. (**D**) Bar chart of MCODE Score of 7 clusters in Cytoscape software. (**E**) Circos plot showing the main functions of *CCL5*, *IL18*, *CXCR6*, *FCER1G*, and *TNFSF13B*, especially in the regulation of immune cells.

**Figure 3 ijms-26-09920-f003:**
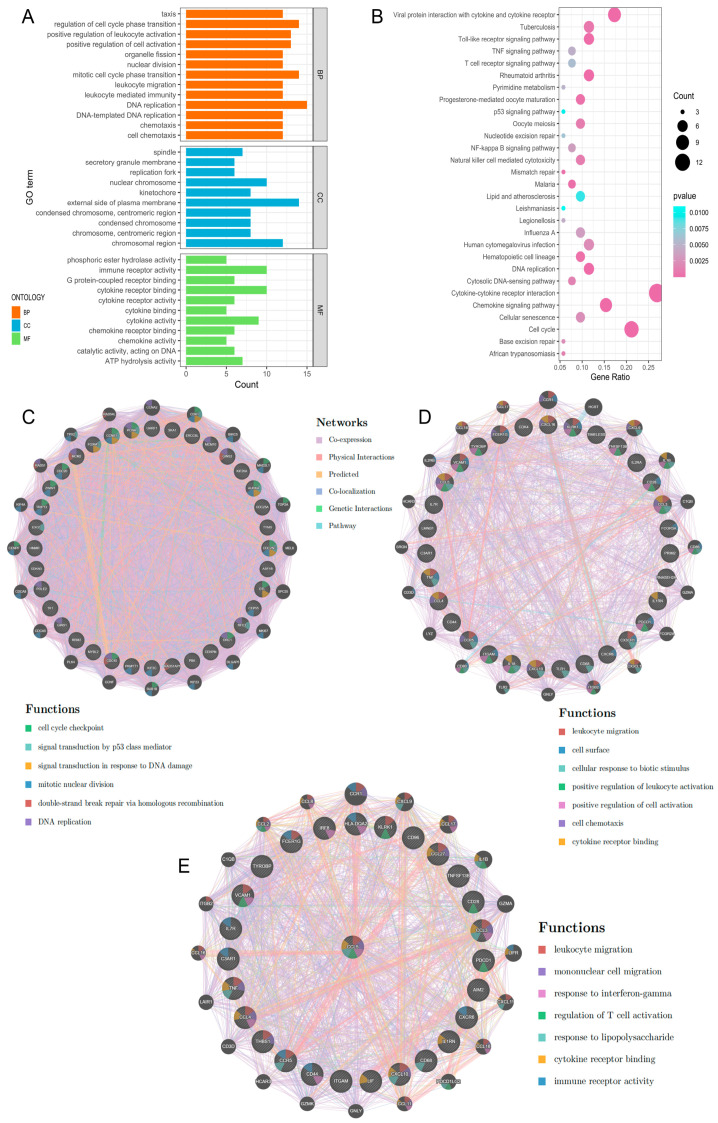
PPI network construction and GO and KEGG pathway analyses of Hub genes. (**A**) The top 10 GO functional enrichment analyses of the hub genes in biological process (BP), cellular component (CC), and molecular function (MF) groups, respectively. (**B**) The top 30 significant KEGG signal pathways of the hub genes. (**C**) The hub genes of Cluster1 with their interaction predicted by PPI network based on GeneMANIA database. The inner circle of PPI network represents hub genes. (**D**) The hub genes of Cluster2 with their interaction predicted by PPI network based on GeneMANIA database. The inner circle of PPI network represents hub genes. (**E**) PPI network of *CCL5* and its first neighbors.

**Figure 4 ijms-26-09920-f004:**
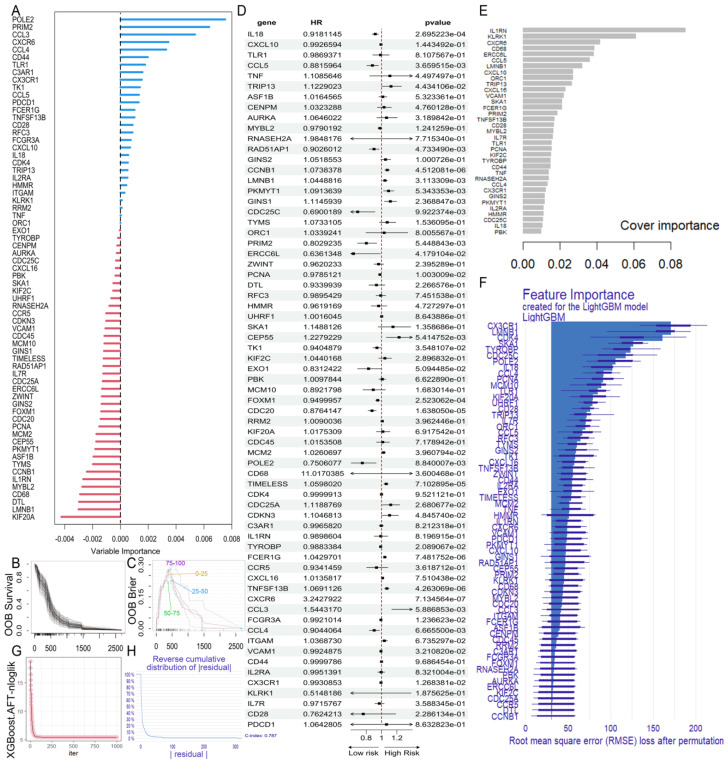
Construction of GBM prognostic risk model. (**A**) The variable importance (VIMP) score and ranking of hub genes in the Random Survival Forest model. (**B**) Survival curve for each individual: the red line represents the overall survival rate. (**C**) Brier score stratified by overall mortality (0–25%, 25–50%, 50–75% and 75–100%): the red line is the overall Brier score. (**D**) Forest plot of the multivariable Cox regression model. (**E**,**F**) Bar chart of VIMP values of XGBoost (with AFT model) and LightGBM. (**G**,**H**) Iterative convergence of the XGBoost (with AFT model) and LightGBM.

**Figure 5 ijms-26-09920-f005:**
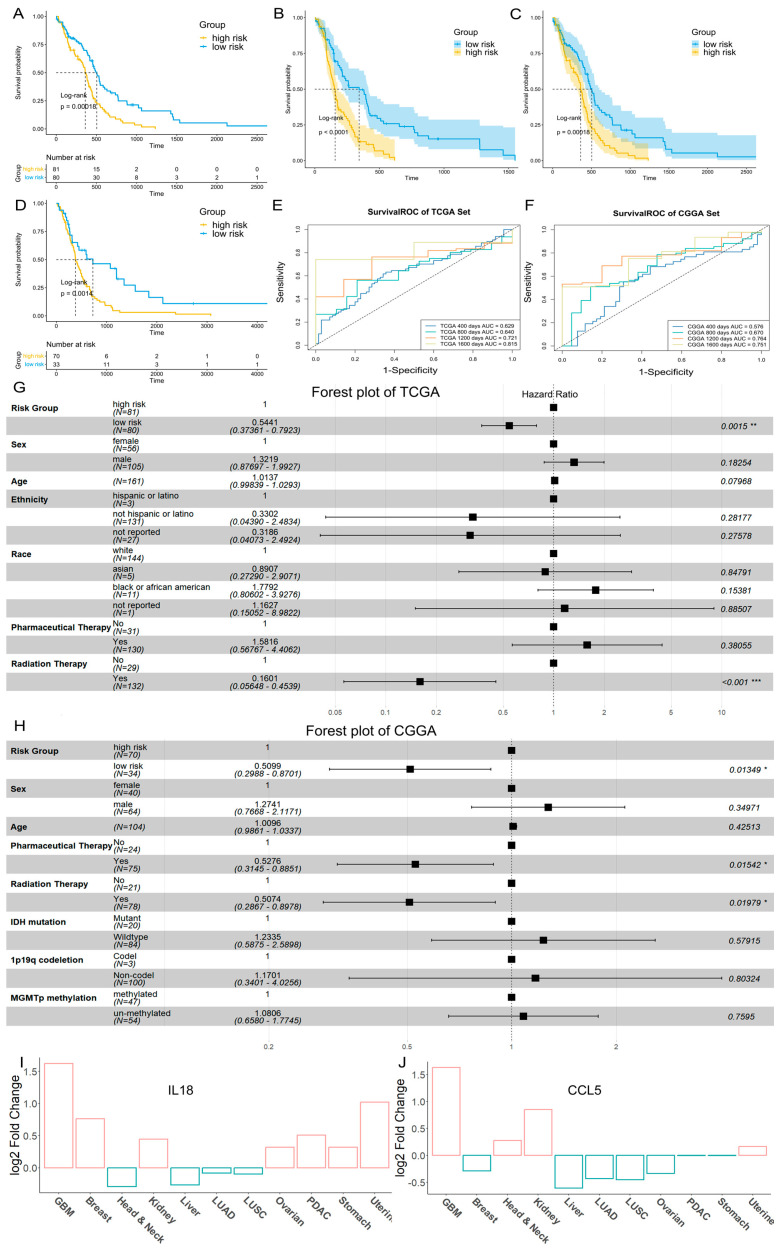
Evaluation and validation of GBM prognostic risk model in TCGA Set and CGGA external validation Set. (**A**) Kaplan–Meier curve in TCGA Set identifying two risk groups. (**B**,**C**) Survival curves for DSS and PFI in TCGA patients with two risk groups. (**D**) Kaplan–Meier curve in CGGA external validation Set using the same criteria of risk. (**E**) Time-dependent Receiver Operating Characteristic (ROC) curve in TCGA Set. (**F**) Time-dependent ROC curve in CGGA external validation Set. (**G**) Forest plot combined with clinical factors and risk group information in TCGA Set. (**H**) Forest plot combined with clinical factors and risk group information in CGGA Set. * indicates significant difference at *p* < 0.05, ** indicates significant difference at *p* < 0.01, *** indicates significant difference at *p* < 0.001. (**I**,**J**) Protein expression of *IL18* and *CCL5* in pan cancer.

**Figure 6 ijms-26-09920-f006:**
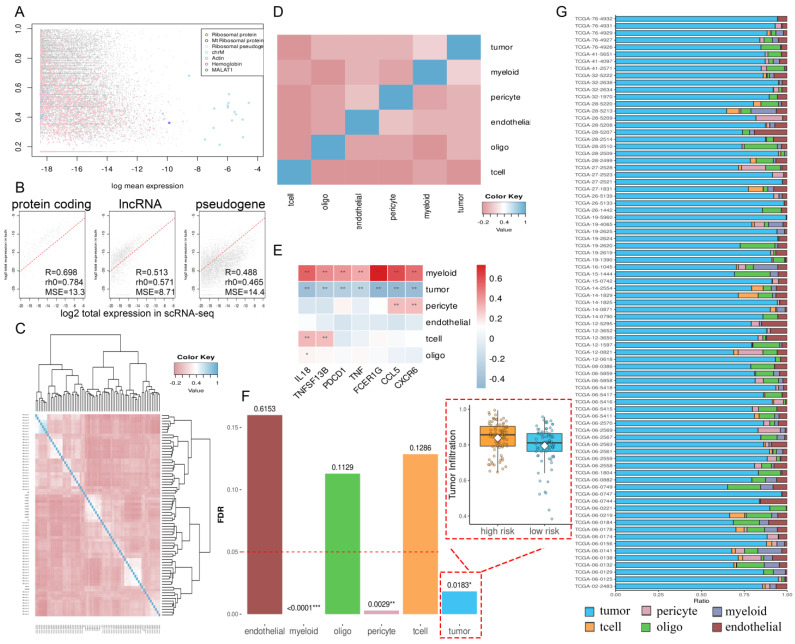
Analysis based on BayesPrism and single-cell RNA-seq data. (**A**) Scatter plot of bulk RNA expression identifying outlier genes. The *x*-axis represents the normalized mean expression of each gene, and the *y*-axis represents its maximum specificity. Different colors represent which type of “outlier” this gene belongs to. (**B**) The consistency of gene expression between TCGA-GBM and single-cell RNA-seq data in terms of mRNA, lncRNA, and pseudogenes. A higher consistency indicates greater reliability of BayesPrism. (**C**) Heatmap showing correlation and clustering between single-cell states. (**D**) Heatmap showing correlation and clustering between single-cell types. (**E**) Heatmap predicting correlation between *IL18*, *CXCR6*, *CCL5*, *FCER1G*, *TNFSF13B*, *PDCD1*, *TNF*, and proportion of six cell types. (**F**) Adjusted *p*-value of differences between high- and low-risk groups in six cell types; below the red line means significant difference. (**G**) Column stacking diagram showing proportion of cells in each GBM patient based on single-cell labels. * indicates significant difference at *p* < 0.05, ** indicates significant difference at *p* < 0.01, *** indicates significant difference at *p* < 0.001.

**Figure 7 ijms-26-09920-f007:**
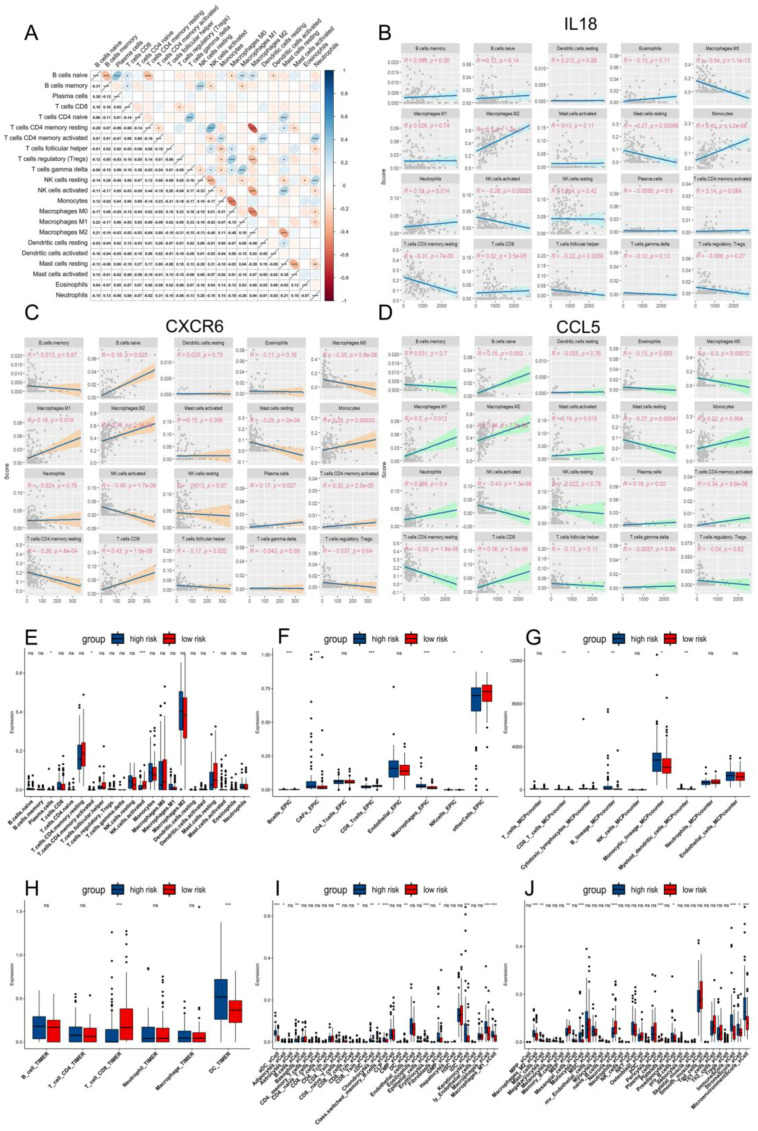
Tumor immune microenvironment in high- and low-risk groups with GBM. (**A**) Correlation heatmap of tumor-infiltrating immune cells using CIBERSORT. (**B**–**D**) The linear relationship between *IL18*, *CXCR6*, *CCL5*, and tumor-infiltrating immune cells; *p*-value < 0.05 indicates significant difference. (**E**–**J**) Most tumor-infiltrating immune cells between high- and low-risk groups had significant differences, such as B cells, CD8^+^ T cells, monocytes, and NK cells, in CIBERSORT, TIMER, EPIC, MCPcounter, and xCell, respectively (the results of xCell included **I** and **J**). ns indicates no significant difference, * indicates significant difference at *p* < 0.05, ** indicates significant difference at *p* < 0.01, *** indicates significant difference at *p* < 0.001.

**Figure 8 ijms-26-09920-f008:**
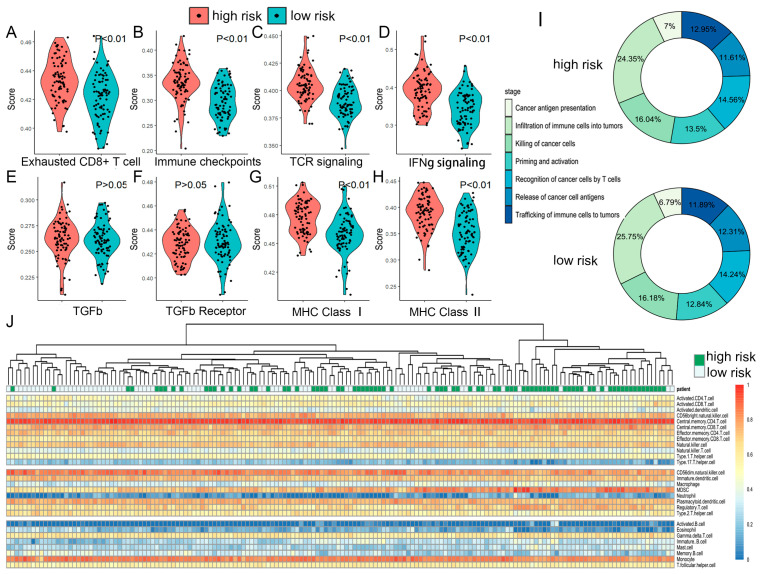
The differences between high and low-risk groups in immune cells, immune signaling pathways, and immune cycle. (**A**–**H**) Violin plot showing the differences in exhaustion-related factors between two risk subgroups. (**I**) The average proportion of high- and low-risk groups in the Cancer-Immunity Cycle (including release of cancer cell antigens, cancer antigen presentation, priming and activation, trafficking of immune cells to tumors, infiltration of immune cells into tumors, recognition of cancer cells by T cells, killing of cancer cells) of Tracking Tumor Immunophenotype (TIP). (**J**) Heatmap showing patients’ clustering based on ssGSEA score. Green represents high-risk group concentrating on the right. White represents high-risk group concentrating on the left.

**Figure 9 ijms-26-09920-f009:**
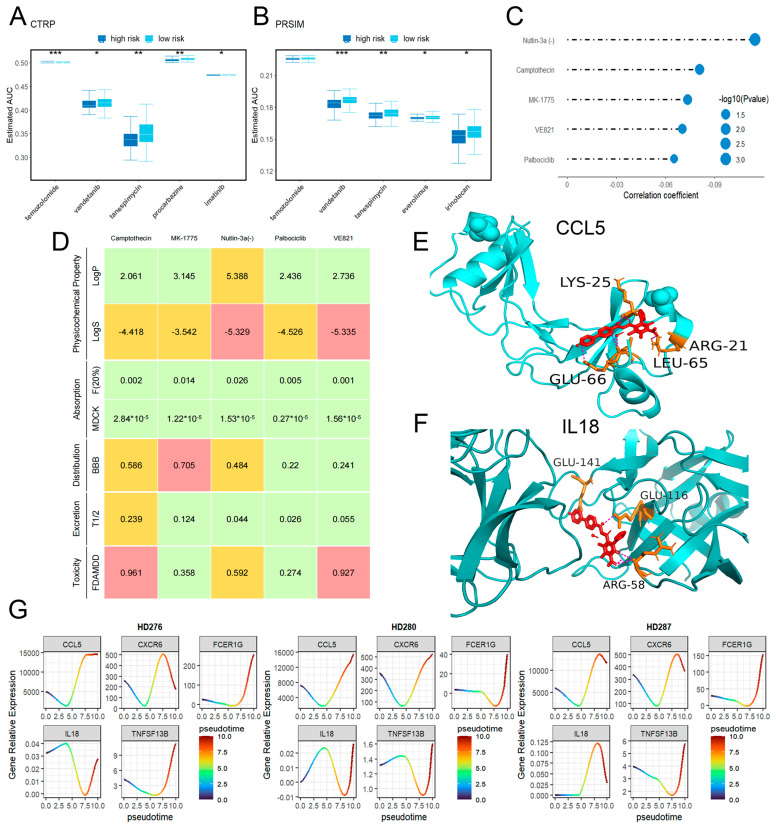
Drug sensitivity and identification of potential therapeutic drugs for key gene targets. (**A**,**B**) Outcomes of variable drug response analysis of common drugs in two pharmacogenomics databases, respectively. (**C**) Identification of potential candidate drugs for core targets in GBM. (**D**) ADMET attributes of potential candidate drugs. (**E**,**F**) Three-dimensional docking model of the core targets (*CCL5* and *IL18*) and Palbociclib; estimated free energy of binding values were −7.06 kcal/mol and −8.89 kcal/mol, respectively. (**G**) Line chart of bulk-pseudotime results demonstrated expression changes in five genes (*CCL5*, *CXCR6*, *FCER1G*, *IL18*, *TNFSF13B*) from resting T cells to exhausted T cells. These genes all increased after depletion began. * indicates significant difference at *p* < 0.05, ** indicates significant difference at *p* < 0.01, *** indicates significant difference at *p* < 0.001.

**Figure 10 ijms-26-09920-f010:**
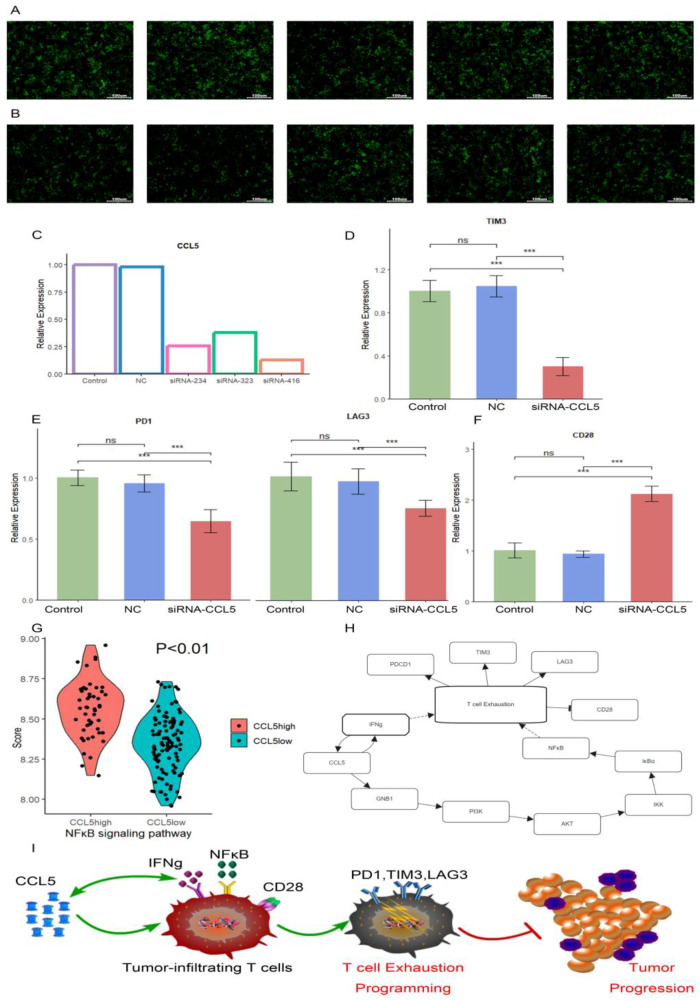
Verification of TEX signature by in vitro experiments and inference of downstream pathways. (**A**,**B**) Image of T cell transfection at 40× magnification for siRNA-416 group and negative control group. Stronger fluorescence indicates higher transfection efficiency. (**C**) Expression of *CCL5* in each group after cell transfection. (**D**,**E**) qRT-PCR was used to detect exhaustion gene expression between the T cells with low *CCL5* expression and cells from two control groups (standard control and negative control); the sample size for each group was 3. (**F**) The expression of *CD28* in the three groups was detected by PCR; the sample size for each group was 3. *** indicates significant difference at *p* < 0.001. (**G**) NFκB signaling pathway in high *CCL5* expression group and low *CCL5* expression group. (**H**) Regulatory network diagram of *CCL5*-induced T cell exhaustion. (**I**) Mechanism diagram of *CCL5*-induced T cell exhaustion.

**Table 1 ijms-26-09920-t001:** Results of Mendelian randomization in IVW.

	Outcome	*β*	Se	*p*-Value
meta-analysis of published GWAS (glioma)				
	CD28^−^ CD8^+^ T cell Absolute Count	−0.0482	0.0299	0.1062
	CD28^−^ CD8^+^ T cell % CD8^+^ T cell	−0.0471	0.0255	0.0652
	CD45RA^−^ CD28^−^ CD8^+^ T cell Absolute Count	−5.3750	3.2070	0.0937
	CD45RA^−^ CD28^−^ CD8^+^ T cell % CD8^+^ T cell	−0.7579	0.3340	0.0233 *
UCSF/Mayo study (GBM)				
	CD28^−^ CD8^+^ T cell Absolute Count	−0.0990	0.0535	0.0642
	CD28^−^ CD8^+^ T cell % CD8^+^ T cell	−0.0964	0.0481	0.0449 *
	CD45RA^−^ CD28^−^ CD8^+^ T cell Absolute Count	−11.9088	5.6194	0.0341 *
	CD45RA^−^ CD28^−^ CD8^+^ T cell % CD8^+^ T cell	−1.0279	0.6422	0.1095
GICC study (GBM)				
	CD28^−^ CD8^+^ T cell Absolute Count	−0.0603	0.0348	0.0833
	CD28^−^ CD8^+^ T cell % CD8^+^ T cell	−0.0547	0.0256	0.0329 *
	CD45RA^−^ CD28^−^ CD8^+^ T cell Absolute Count	−4.2093	3.1080	0.1756
	CD45RA^−^ CD28^−^ CD8^+^ T cell % CD8^+^ T cell	−0.3285	0.3367	0.3293

* indicates significant difference at *p* < 0.05.

**Table 2 ijms-26-09920-t002:** Information of the final five core genes in multivariable Cox regression.

Gene	Ensembl IDs	Location	Hazard Ratios	95% CI	*p*-Value
*IL18*	ENSG00000150782	Chromosome 11: 112,143,253–112,164,096 reverse strand	0.9181	0.8769, 0.9613	0.0003
*CXCR6*	ENSG00000172215	Chromosome 3: 45,940,933–45,948,351 forward strand	3.2428	2.0367, 5.1630	*p* < 0.0001
*CCL5*	ENSG00000271503	Chromosome 17: 35,871,491–35,880,793 reverse strand	0.8816	0.8098, 0.9598	0.0037
*FCER1G*	ENSG00000158869	Chromosome 1: 161,215,234–161,220,699 forward strand	1.0430	1.0239, 1.0623	*p* < 0.0001
*TNFSF13B*	ENSG00000102524	Chromosome 13: 108,251,240–108,308,484 forward strand	1.0691	1.0391, 1.1000	*p* < 0.0001

**Table 3 ijms-26-09920-t003:** Data sources.

Data Name	Database	Type	Detail	Data
Meta-analysis of published GWAS about glioma	NHGRI-EBI GWAS catalog	GWAS	A meta-analysis of published GWAS covering phenotypes of non-glioblastoma glioma, glioma, and glioma (high-grade) [61]	28 March 2024
GICC study (GBM)	Glioma International Case Control Consortium	GWAS	A GWAS of 4572 cases and 3286 controls performed by the Glioma International Case Control Consortium [62]	
UCSF/Mayo study (GBM)	University of California, San Francisco (UCSF)-Mayo	GWAS	A GWAS of 1591 cases and 804 controls from the University of California, San Francisco (UCSF)-Mayo [62]	
Public GWAS related to TEX	MR Base GWAS catalog	GWAS	A report about 731 immune cell traits in a cohort of 3757 Sardinians [63] (GCST90001686,GCST90001687,GCST90001695,GCST90001696)	28 March 2024
TCGA-GBM	TCGA	bulk RNA-seq	The project of The Cancer Genome Atlas included 167 GBM patients	15 November 2023
GSE234100	GEO	bulk RNA-seq	Primary human T cells from three healthy donors were TCR-transduced and stimulated with cognate antigen (NY-ESO-1) to generate effector cells (TEFF, 1× stimulation) and exhausted cells (TEX, 4× stimulation) [19]	1 July 2023
GSE103224	GEO	single-cell RNA-seq	Performed single-cell RNA-seq on tens of thousands of dissociated high-grade glioma tissue cells from 8 human patients [68]	2 July 2018
GSE210534	GEO	bulk RNA-seq	Four human healthy donor T cells were isolated, transduced with an NY-ESO-1 TCR lentivirus construct, stimulated in four different conditions (Trested, Ttumor, TEX, Teff) [56]	7 November 2022
CGGA.mRNAseq_325.ReadCounts-genes	CGGA	bulk RNA-seq	The first batch of sequencing data released by Chinese Glioma Genome Atlas includes 325 samples from Chinese cohort [64,65,66]	20 June 2022

## Data Availability

The datasets supporting the conclusions of this article are included within the article and Table 3. The experimental results have been uploaded separately.

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
