# Peer review of "Multiomics Investigation of Exhausted T Cells in Glioblastoma Tumor Microenvironment: CCL5 as a Prognostic and Therapeutic Target"

_ijms, 2025, doi:10.3390/ijms26209920_

Round 1

Reviewer 1 Report

Comments and Suggestions for Authors

This manuscript by Qin et al. applies multi-omics approaches to study exhausted T cells in GBM.  It proposes CCL5 as both a prognostic biomarker and therapeutic target. The authors integrate available bulk transcriptomics, single-cell data, drug screens, and limited in vitro validation. While the topic is timely and of potential interest to the tumor immunology and neuro-oncology fields, the manuscript is difficult to follow, raises concerns about overreliance on computational pipelines, and lacks sufficient wet-lab validation to support its conclusions.

Impact: The idea of dissecting T cell exhaustion in glioblastoma is important, as immunotherapy strategies in this disease have been largely unsuccessful. Identifying actionable drivers of exhaustion could, in principle, open new therapeutic avenues. However, the current manuscript falls short of making a convincing impact. The narrative is convoluted, with extensive results from diverse bioinformatics analyses but little clarity on the biological logic tying them together. The reliance on in silico analyses without rigorous experimental validation leaves major uncertainty about the robustness of the findings. The limited qPCR assays are insufficient to support the broad mechanistic claims. At times, the writing style and odd result selection raise concerns about whether generative AI was used in manuscript drafting, and whether some results may be computational artifacts or “hallucinations.”

Major concerns:

  1. Clarity and Rigor: The manuscript is very difficult to follow. Results are presented in long, unfocused blocks with limited interpretation. It reads more like a data dump than a carefully reasoned scientific story.

  2. Possible AI Generation: Portions of the text are oddly phrased, repetitive, and contain sweeping conclusions unsupported by data. This raises concern about generative AI use in manuscript preparation.

  3. Insufficient Validation: Virtually all results are computational. The only experimental validation consists of knockdown and qPCR assays, which is far too limited to substantiate the wide-ranging claims about GBM prognosis, immune biology, and therapeutic targeting.

  4. Overinterpretation of Computational Results: Many associations (e.g., Mendelian randomization, pseudotime analyses) are taken at face value as causal, but these methods have substantial caveats. The manuscript does not adequately acknowledge limitations.

  5. Therapeutic Implications Are Premature: The suggestion of Palbociclib as a GBM therapy based solely on docking simulations and risk models is speculative and not justified by the data.

  6. Reproducibility: Methods are not described with enough clarity to ensure reproducibility. The flow of data from dataset to conclusion is often opaque.  ----

  1. Minor concerns:
  2. The introduction contains excessive background without clear articulation of the knowledge gap this study fills.

  3. Numerous grammatical errors and awkward phrasing reduce readability.

  4. The discussion is overly speculative and not well balanced with limitations.

  5. PCV is not a major treatment option for GBM (maybe for grade 2 and 3 gliomas, not really for GBM).

  6. PCV does not include everolimus as stated in the end of Results section

  7. Define TEX in the abstract

Comments on the Quality of English Language

see above about clarity and at times grammar.

Author Response

Dear Reviewer 1,

Thank you very much for your comments regarding our manuscript entitled Multiomics Investigation of Exhausted T cells in Glioblastoma Tumor Microenvironment: CCL5 as a Prognostic and Therapeutic Target [Manuscript ID: ijms-3884261].

Thank you for your letter dated Sep 10. We are truly grateful to your critical comments and thoughtful suggestions on how to improve our manuscript.

The valuable comments and criticisms have provided important guidance for our research and the revision of our manuscript. Based on these comments and suggestions, we have made appropriate modifications on the original manuscript, then we hope to meet your approval. In the revised manuscript, all modifications have been highlighted in red. The reviewer’s comments are reproduced, and our responses are provided directly below each comment in blue. The corresponding revisions are addressed point by point in the main text.

Once again, we deeply appreciate the time and effort dedicated by you. All revisions have been carefully implemented as highlighted in the revised manuscript (marked in red).

Sincerely,

Lei Cao

Overall comments:

This manuscript by Qin et al. applies multi-omics approaches to study exhausted T cells in GBM.  It proposes CCL5 as both a prognostic biomarker and therapeutic target. The authors integrate available bulk transcriptomics, single-cell data, drug screens, and limited in vitro validation. While the topic is timely and of potential interest to the tumor immunology and neuro-oncology fields, the manuscript is difficult to follow, raises concerns about overreliance on computational pipelines, and lacks sufficient wet-lab validation to support its conclusions.

Impact: The idea of dissecting T cell exhaustion in glioblastoma is important, as immunotherapy strategies in this disease have been largely unsuccessful. Identifying actionable drivers of exhaustion could, in principle, open new therapeutic avenues. However, the current manuscript falls short of making a convincing impact. The narrative is convoluted, with extensive results from diverse bioinformatics analyses but little clarity on the biological logic tying them together. The reliance on in silico analyses without rigorous experimental validation leaves major uncertainty about the robustness of the findings. The limited qPCR assays are insufficient to support the broad mechanistic claims. At times, the writing style and odd result selection raise concerns about whether generative AI was used in manuscript drafting, and whether some results may be computational artifacts or “hallucinations.”

Response: We would like to thank you for your careful reading, helpful comments, and constructive suggestions, which has significantly improved the presentation of our manuscript.

Major concerns:

Major concerns 1: Clarity and Rigor: The manuscript is very difficult to follow. Results are presented in long, unfocused blocks with limited interpretation. It reads more like a data dump than a carefully reasoned scientific story.

Response 1: Thank you for the suggestion to enhance our explanations. To improve the clarity and rigor of our manuscript, we have undertook the following revisions.

We have revised the Introduction section to clarify the overall workflow of our study, and improved the expression and logic to highlight the knowledge gap this study filled. For example, we have revised the description of workflow on Page 2 Lines 67-72 in the revised manuscript.

We have polished the text in Results section to emphasize the our findings and removed unnecessary sentences. For example, we have polished our text on Page 12 Lines 296-299.

We have strengthened the connections between sentences and paragraphs to ensure a clearer and more coherent expression. For example, we have added a sentence on Page 12 Lines 258-259, to provide a clearer link between the two subsections.

We have also revised the Discussion section to enhance logic and rigor. For example, we have added the descriptions of limitations.

Major concerns 2: Possible AI Generation: Portions of the text are oddly phrased, repetitive, and contain sweeping conclusions unsupported by data. This raises concern about generative AI use in manuscript preparation.

Response 2: We sincerely appreciate your suggestion on improving our text. We sincerely apologize for the our poor language quality and guarante that the content of our manuscript is not generated by AI.

As you suggested, we have carefully revised the manuscript to remove repetitive or unclear expressions and enhanced the overall language quality. Here are some specific examples of our revisions:

On Page 8, Line 194:

- Original: “This core gene set (CXCR6, IL18, FCER1G, CCL5, TNFSF13B) was utilized as the optimal gene combination to construct a prognostic risk model.”

- Revised: “These key genes were utilized to construct a risk model.”

On Page 10, Lines 242-246:

- Original: “IL18 and CCL5 proteins showed pronounced overexpression in gliomas (Figure 5I,J), while FCER1G also exhibited increased expression in brain cancer (Figure S1d). Notably, while CCL5 expression is downregulated across most tumor types, it demonstrates the highest expression levels in brain tumors, potentially highlighting its unique role in gliomas progression. Furthermore, FCER1G displayed a strong correlation with IL18 (Figure S1e), suggesting potential functional relationships.”

- Revised: “Pan-cancer analysis showed that IL18 and CCL5 proteins were overexpression in GBM compared with other tumor (Figure 5I,J). FCER1G also showed increased expression (Figure S2d). Notably, while CCL5 expression was downregulated in most tumor types, it showed the highest expression levels in GBM, indicating its unique role in GBM progression.”

On Page 20, Lines 473-476:

- Original: “Utilizing these biomarkers, we stratified GBM patients into high- and low-risk groups based on the extent of T cell exhaustion.”

- Revised: “Utilizing these key genes, we constructed a reliable prognostic model and stratified GBM patients into high- and low-risk groups of T cell exhaustion.”

On Page 20, Lines 473-477:

- Original: “Our proteomic analysis further validated the overexpression of CCL5, IL18, and FCER1G in GBM tissues compared to normal tissues, with CCL5 and IL18 in particular showing significantly higher levels in brain tumors. The differential expression of CCL5 suggests a unique role for this chemokine in GBM, as it is known to be involved in immune cell recruitment and activation.”

- Revised: “Our proteomic analysis confirmed the overexpression of CCL5, IL18, and FCER1G in GBM compared to normal tissues. Notably, CCL5 showed significantly higher levels in brain tumors, suggestting a unique role in GBM.”

Major concerns 3: Insufficient Validation: Virtually all results are computational. The only experimental validation consists of knockdown and qPCR assays, which is far too limited to substantiate the wide-ranging claims about GBM prognosis, immune biology, and therapeutic targeting.

Response 3: Thanks for pointing out our shortcomings. We acknowledge that our experimental validation was primarily based on RT-qPCR analysis, without further validation.

As this study was our first work on GBM, we paid more attentions to exploratory research. The main purpose of our study was to identify potential key genes and explore their immunological significance through multiomics analysis. We also analysed the immune tumor microenvironment and drug sensitivity of GBM, providing a reference for further research or clinical practice.

To enhance the robustness of our conclusions, we not only utilized large-scale data from TCGA database, but also validated in CGGA data, an external cohort from China. The results from these datasets were highly consistent with our analysis. In bioinformatic analysis, we selected appropriate methods. For example, we used the exactTest (from “edgeR” R package) for data with small sample sizes, in order to reduce bias. Our bioinformatic analysis provided additional insights while exploring and validating the role of biomarkers, which pointed to the direction of our future research. For example, we found that these biomarkers may be involved in cell communication between T cells, macrophages, and microglia, which was meaningful for further exploring the tumor microenvironment.

Our in in vitro experiments were designed to provide preliminary evidence for the rationale and feasibility of targeting CCL5. We chose RT-qPCR, because it had high sensitivity, specificity, and reproducibility, making it the good standard for gene expression analysis and validating transcriptome datasets[1-4]. But due to limitations in experimental conditions and facilities, we were unable to perform further validation experiments, which we have addressed as a limitation in the Discussion section. In the future, we plan to perform additional experiments, such as establishing animal models, to further investigate T cell exhaustion in GBM.

We will further investigate the underlying biological mechanisms in the future through more analysis and experiments. And we clearly acknowledge in the Discussion (Page 21 Lines 519-521 and Page 22 Lines 543-549) that the current experimental validation was limited.

Furthermore, we have refined the language and improved the logical flow to ensure the rigor of our conclusions.

Major concerns 4: Overinterpretation of Computational Results: Many associations (e.g., Mendelian randomization, pseudotime analyses) are taken at face value as causal, but these methods have substantial caveats. The manuscript does not adequately acknowledge limitations.

Response 4: We sincerely appreciate your suggestion on limitations of our manuscript. We have added discussions of the limitations of Mendelian randomization and pseudotime analysis in the Discussion section.

Mendelian Randomization (MR) is a method used to assess causal relationship. However, the ability of MR to validate time relationships is limited, which means MR may not fully capture the true causal relationships. In addition, MR also has issues such as weak instrumental bias, horizontal pleiotropy and heterogeneity. To address potential issues in MR, we performed screening (F-statistic, minimum linkage disequilibrium (LD) R-squared value and minor allele frequency (MAF), Page 23 Lines 560-564 and Page 23 Lines 575-576) and testing (MR-Egger and Cochran Q test, Page 23 Lines 571-574), which could enhance the reliability of our results to some extent. However, Mendelian randomization cannot verify the time relationships in causal. According to the definition of exhausted T cells and current research results, T cell exhaustion generally occurs after disease and persistent antigen stimulation[5-7]. Therefore, we considered this widely accepted conclusion and employed a two-sample Mendelian Randomization to identify potential causal relationship. We have revised the contents of the Mendelian randomization in the Discussion (Page 20 Lines 458-460).

Pseudotime analysis could reflect relative ordering to a certain extent, but it does not represent actual biological time. For the five key genes (CCL5, IL18, CXCR6, FCER1G, TNFSF13B), we used pseudotime analysis to determine the most likely and promising gene for our subsequent experiments. However, the potential roles of other genes may still be overlooked, and we have acknowledged this limitation on Page 21 Lines 509-511 in the revised manuscript.

Major concerns 5: Therapeutic Implications Are Premature: The suggestion of Palbociclib as a GBM therapy based solely on docking simulations and risk models is speculative and not justified by the data.

Response 5: Thank you for your interest in the candidate drug analysis. We acknowledge that we did not investigate the binding capacity between candidate drug and our potential targets through more experiments. But our analysis referred to relevant studies. These studies identified candidate drugs using pharmacogenomic data and subsequently validated their feasibility through molecular docking models.

For example, Shuang Guo identified potential candidate drugs for ovarian cancer through pharmacogenomic analysis, obtained their pharmacokinetic characteristics using ADMET, and finally performed AutoDock analysis to ensure the interactions between the identified drugs and their targets[8]. Run Shi also used pharmacogenomic analysis to explore potential drugs for treating bladder cancer[9]. Run Shi’s study set a filtering threshold of negative r value and p value less than 0.05, which was same as ours[9]. Therefore, in theory, our analysis were feasible. Based on these analysis, we have also found some relevant researches indicating that Palbociclib was used to treat brain tumors originating from breast cancer[10, 11]. This suggested that Palbociclib could cross the blood-brain barrier, thereby enhancing the reliability of our conclusion.

The above content had been described in original manuscript, and in order to express ourselves clearly, we have made some modifications on Pages 15-16 Lines 347-348, Page 16 Lines 355-359 and Page 21 496-499. We have also added this limitation on Page 21 Lines 499-502 this limitation in the revised manuscript. Your suggestion is highly pertinent, and we will explore the establishment of animal models in future studies to find more evidence.

Major concerns 6: Reproducibility: Methods are not described with enough clarity to ensure reproducibility. The flow of data from dataset to conclusion is often opaque. 

Response 6: We appreciate your suggestion regarding unclear description of our Methods. We acknowledge that our previous description was insufficient.

For public datasets, we have indicated their sources and provided a brief description in Table 3 (Page 23). We also added descriptions of the statistical methods and the R packages used in the Methods section. Here are some specific examples of our revisions:

On Page 24, Lines 604-606:

- Added: “The exactTest function (from “edgeR” package) was used for comparison between groups, which was suitable for small sample dataset. P-values were adjusted for multiple comparisons using the False Discovery Rate (FDR) method.”

On Page 24, Lines 633-636:

- Added: “Prior to model construction, we performed necessary data preprocessing. For patient mortality outcomes, we utilized available death dates, setting the survival status to 1 when present. In cases where the death date was missing or unclear, we imputed using the date of last follow-up.”

On Page 25, Lines 652-654:

- Added: “The decision curve analysis (DCA) was performed by the “ggDCA” R packages to evaluate the clinical utility of survival model.”

To enhance the transparency of our results, we have uploaded the drug sensitivity analysis outcomes and the molecular docking files of AutoDock (both pdb format and pdbqt format) in supplementary materials (Supplementary DS.zip: Data of drug sensitivity; Supplementary AutoDock.zip: molecular docking models of CCL5 and IL18;). These files could further improved the credibility of our study.

Minor concerns:

Minor concerns 1: The introduction contains excessive background without clear articulation of the knowledge gap this study fills.

Response 7: Thank you for your suggestion regarding introduction. We acknowledge that our previous description was unclear.

We have revised the Introduction to improve the logic. Specifically, we have moved the text describing the knowledge gap that our study seeks to fill to the end of the second paragraph (Page 2 Lines 62-64).

In addition, we have revised other parts of the Introduction. Here are some specific examples of our revisions:

On Page 1, Lines 36-37:

- Original: “Exhausted T (TEX) cells are a subset of T lymphocytes characterized by features of cellular senescence.”

- Revised: “Exhausted T (TEX) cells represent a subset of T lymphocytes characterized by progressive loss of effector functions.”

On Page 2, Lines 50-52:

- Original: “However, recent research has dramatically altered this perspective, revealing a close interaction between the brain and its unique immune microenvironment.”

- Revised: “However, recent research has fundamentally changed the view of brain immunity, revealing its close interaction with the immune system and the presence of a unique immune microenvironment.”

To improve readability, we have also adjusted the order of some sentences. All modifications have been highlighted in the revised manuscript.

Minor concerns 2: Numerous grammatical errors and awkward phrasing reduce readability.

Response 8: Thank you very much for your advice. We have corrected the spelling errors and grammatical errors in the manuscript, especially the use of modal verbs. Here are some specific examples of our revisions:

On Page 2, Lines 50-52:

- Original: “However, recent research has dramatically altered this perspective, revealing a close interaction between the brain and its unique immune microenvironment.”

- Revised: “However, recent research has fundamentally changed the view of brain immunity, revealing its close interaction with the immune system and the presence of a unique immune microenvironment.”

On Page 6, Lines 150-151:

- Original: “They may lead to T cell exhaustion by reducing the cell cycle signatures and p53 signaling transduction.”

- Revised: “They might lead to T cell exhaustion by reducing the cell cycle signatures and p53 signaling transduction.”

On Pages 15-16, Lines 347-348:

- Original: “We then conducted ADMET analysis of these candidate drugs using ADMETlab2.0”

- Revised: “We then analyzed pharmacokinetic characteristics of these candidate drugs using ADMETlab2.0”

On Page 20, Lines 454-456:

- Original: “Notably, our in silico and cell experiments provide novel evidence that CCL5, one of our identified biomarkers, binds to the proposed drug candidate and is involved in the regulation of T cell exhaustion.”

- Revised: “Notably, our in silico and cell experiments provided novel evidence that CCL5, one of our identified key genes, bound to the proposed candidate drug and was involved in the regulation of T cell exhaustion.”

Minor concerns 3: The discussion is overly speculative and not well balanced with limitations.

Response 9: We appreciate your suggestion regarding Discussion.

We acknowledged certain shortcomings in the experimental, but the discussion of other negative results was insufficient, including The issues you mentioned above. We have revised the Discussion and added additional content.

The modifications were added on Page 20 Lines 459-460, Page 20 Lines 475-477, Page 21 Lines 487-488, Page 21 Lines 499-502, Page 21 Lines 509-511 and Page 22 Lines 543-549.

Minor concerns 4: PCV is not a major treatment option for GBM (maybe for grade 2 and 3 gliomas, not really for GBM).

Response 10: Thanks for this helpful comment! We apologize for not expressing ourselves clearly. According to guidelines from the United States and China[12, 13], TMZ is recognized first-line chemotherapy drug. However, when TMZ is resistant, unavailable, or in cases of GBM recurrence or progression, some hospitals or centers will used PCV as an alternative chemotherapy or combination therapy. Khasraw’s study used a combination therapy of temozolomide and procarbazine to treat a specific subtype of GBM[14]. Team of Se-Hyuk Kim explored the effects of Procarbazine and CCNU chemotherapy[15]. There were also studies that have explored the efficacy of PCV in treating recurrent GBM[16, 17]. Some researches showed that everolimus could prolong progression-free survival (PFS) in patients with newly diagnosed GBM[18]. Andrew’ research also reported a case about successful treatment of a TSC2-mutant glioblastoma with everolimus[19]. Therefore, we have also selected PCV and other second-line agents in this article as treatment option, in order to provide a reference for other researchers.

Minor concerns 5: PCV does not include everolimus as stated in the end of Results section.

Response 11: We thank the reviewer for this important point. We apologize for the confusion caused by the our text. In original manuscript, we wanted to discuss PCV and everolimus separately, rather than to express that everolimus is included in the PCV. As our response in minor concern 4, Some researches showed that everolimus could prolong progression-free survival (PFS) in patients with newly diagnosed GBM[18], and Andrew’ research also reported a case about successful treatment of a TSC2-mutant glioblastoma with everolimus[19]. Therefore, we considered everolimus in our analysis.

We have revised this sentence to better describe our results. The modifications were highlighted on Page 15 Lines 339-340 in the revised manuscript.

Minor concerns 6: Define TEX in the abstract.

Response 12: Thank you for this helpful comment. We apologize for not expressing TEX in the abstract clearly.

As you suggested, we have added a brief definition of TEX on Page 1 Abstract Lines 13-14 in the revised manuscript.

References:

  1. de Brito MWD, de Carvalho SS, Mota MBdS, Mesquita RD. RNA-seq validation: software for selection of reference and variable candidate genes for RT-qPCR. BMC Genomics, 2024,25(1), 697.
  2. Bustin S. Quantification of mRNA using real-time reverse transcription PCR (RT-PCR): trends and problems. Journal of Molecular Endocrinology, 2002,29(1), 23-39.
  3. Liang L, He Z, Yu H, Wang E, Zhang X, Zhang B, Zhang C, Liang Z. Selection and Validation of Reference Genes for Gene Expression Studies in Codonopsis pilosula Based on Transcriptome Sequence Data. Scientific Reports, 2020,10(1), 1362.
  4. Chung Y-S, Lee N-J, Woo SH, Kim J-M, Kim HM, Jo HJ, Park YE, Han M-G. Validation of real-time RT-PCR for detection of SARS-CoV-2 in the early stages of the COVID-19 outbreak in the Republic of Korea. Scientific Reports, 2021,11(1), 14817.
  5. Blank CU, Haining WN, Held W, Hogan PG, Kallies A, Lugli E, Lynn RC, Philip M, Rao A, Restifo NP, Schietinger A, Schumacher TN, Schwartzberg PL, Sharpe AH, Speiser DE, Wherry EJ, Youngblood BA, Zehn D. Defining 'T cell exhaustion'. Nat Rev Immunol, 2019,19(11), 665-674.
  6. Wherry EJ. T cell exhaustion. Nat Immunol, 2011,12(6), 492-9.
  7. Chu T, Wu M, Hoellbacher B, de Almeida GP, Wurmser C, Berner J, Donhauser LV, Ann-Katrin G, Lin S, Cepeda-Mayorga JD, Kilb, II, Bongers L, Toppeta F, Strobl P, Youngblood B, Schulz AM, Zippelius A, Knolle PA, Heinig M, Hackstein CP, Zehn D. Precursors of exhausted T cells are preemptively formed in acute infection. Nature, 2025.
  8. Guo S, Liu Y, Sun Y, Zhou H, Gao Y, Wang P, Zhi H, Zhang Y, Gan J, Ning S. Metabolic-Related Gene Prognostic Index for Predicting Prognosis, Immunotherapy Response, and Candidate Drugs in Ovarian Cancer. J Chem Inf Model, 2024,64(3), 1066-1080.
  9. Shi R, Wang X, Wu Y, Xu B, Zhao T, Trapp C, Wang X, Unger K, Zhou C, Lu S, Buchner A, Schulz GB, Cao F, Belka C, Su C, Li M, Shu Y. APOBEC-mediated mutagenesis is a favorable predictor of prognosis and immunotherapy for bladder cancer patients: evidence from pan-cancer analysis and multiple databases. Theranostics, 2022,12(9), 4181-4199.
  10. Abusanad A, Al Hashem H. A Substantial Response from Adding Palbociclib to Endocrine Therapy in Brain Metastasis from Hormone Receptor-Positive, HER2-Negative Breast Cancer: Case Reports. Case Rep Oncol, 2021,14(1), 446-452.
  11. Steiger HJ, Vollmer K, Rogers S, Schwyzer L. State of affairs regarding targeted pharmacological therapy of cancers metastasized to the brain. Neurosurg Rev, 2022,45(5), 3119-3138.
  12. Horbinski C, Nabors LB, Portnow J, Baehring J, Bhatia A, Bloch O, Brem S, Butowski N, Cannon DM, Chao S, Chheda MG, Fabiano AJ, Forsyth P, Gigilio P, Hattangadi-Gluth J, Holdhoff M, Junck L, Kaley T, Merrell R, Mrugala MM, Nagpal S, Nedzi LA, Nevel K, Nghiemphu PL, Parney I, Patel TR, Peters K, Puduvalli VK, Rockhill J, Rusthoven C, Shonka N, Swinnen LJ, Weiss S, Wen PY, Willmarth NE, Bergman MA, Darlow S. NCCN Guidelines(R) Insights: Central Nervous System Cancers, Version 2.2022. J Natl Compr Canc Netw, 2023,21(1), 12-20.
  13. Zhao MJ, Lu T, Ma C, Wang ZF, Li ZQ. A narrative review on the management of glioblastoma in China. Chin Clin Oncol, 2022,11(4), 29.
  14. Khasraw M, Lee A, McCowatt S, Kerestes Z, Buyse ME, Back M, Kichenadasse G, Ackland S, Wheeler H. Cilengitide with metronomic temozolomide, procarbazine, and standard radiotherapy in patients with glioblastoma and unmethylated MGMT gene promoter in ExCentric, an open-label phase II trial. J Neurooncol, 2016,128(1), 163-171.
  15. Kim S-H, Yoo H, Chang JH, Kim C-Y, Chung DS, Kim SH, Park S-H, Lee YS, Yang SH. Procarbazine and CCNU Chemotherapy for Recurrent Glioblastoma with MGMT Promoter Methylation. Journal of Korean Medical Science, 2018,33(24).
  16. Weller M, Le Rhun E. How did lomustine become standard of care in recurrent glioblastoma? Cancer Treat Rev, 2020,87, 102029.
  17. Brada M, Stenning S, Gabe R, Thompson LC, Levy D, Rampling R, Erridge S, Saran F, Gattamaneni R, Hopkins K, Beall S, Collins VP, Lee SM. Temozolomide versus procarbazine, lomustine, and vincristine in recurrent high-grade glioma. J Clin Oncol, 2010,28(30), 4601-8.
  18. Pinto-Fraga J, Garcia-Chico C, Lista S, Lacal PM, Carpenzano G, Salvati M, Santos-Lozano A, Graziani G, Ceci C. Protein kinase inhibitors as targeted therapy for glioblastoma: a meta-analysis of randomized controlled clinical trials. Pharmacol Res, 2025,212, 107528.
  19. Zureick AH, McFadden KA, Mody R, Koschmann C. Successful treatment of a TSC2-mutant glioblastoma with everolimus. BMJ Case Rep, 2019,12(5).

Reviewer 2 Report

Comments and Suggestions for Authors

  1. The sample size of the data is very small with respect to the TEX and TEFF datasets. The author needs to increase the sample size to minimize the outlier effect or batch variation. Single cell data from only 8 patients is not enough, though the cell number is in a good range. The tumor microenvironment cannot be studied completely.
  2. The study mostly relies on the bioinformatic predictions, which need a huge dataset and a large sample size. The multiomics data are impressive but rely on in-silico modeling. Most wet lab experiments are needed to strengthen the link between CCL5 and TEX.
  3. The paper lacks the mechanistic depth to conclude the CCL5 and TEX via NFkB signaling. The author should perform the experiment related to phosphorylation or NfkB reporter activity to directly correlate in T cells, without CCL5 knockdown. Also, the author can check the activity of the exhaustion markers by treating CCL5 high T cells with NFkB inhibitors. There are various experiments that need to be included in this manuscript to conclude CCL5 as a Prognostic and Therapeutic Target.
  4. The various models' data included in the paper seem to be overfitting, though the risk model used many approaches. The clinical utility of this is questionable, and the author should include the decision curve analysis for better understanding.
  5. Single-cell pictures are very blurred. Please try to get more focused pictures.

Author Response

Dear Reviewer 2,

Thank you very much for your comments regarding our manuscript entitled Multiomics Investigation of Exhausted T cells in Glioblastoma Tumor Microenvironment: CCL5 as a Prognostic and Therapeutic Target [Manuscript ID: ijms-3884261].

Thank you for your letter dated Sep 10. We are truly grateful to your critical comments and thoughtful suggestions on how to improve our manuscript.

The valuable comments and criticisms have provided important guidance for our research and the revision of our manuscript. Based on these comments and suggestions, we have made appropriate modifications on the original manuscript, then we hope to meet your approval. In the revised manuscript, all modifications have been highlighted in red. The reviewer’s comments are reproduced, and our responses are provided directly below each comment in blue. The corresponding revisions are addressed point by point in the main text.

Once again, we deeply appreciate the time and effort dedicated by you. All revisions have been carefully implemented as highlighted in the revised manuscript (marked in red).

Sincerely,

Lei Cao

Comment 1: The sample size of the data is very small with respect to the TEX and TEFF datasets. The author needs to increase the sample size to minimize the outlier effect or batch variation. Single cell data from only 8 patients is not enough, though the cell number is in a good range. The tumor microenvironment cannot be studied completely.

Response 1: Thank you for your attention to the sample size of the data.

The TEX and TEFF datasets were obtained from the GEO database (GSE234100). GSE234100 is one of the few datasets that sequenced pure cell line of human exhausted T cells[1]. And this datasets is internally comparable and of high quality. Nevertheless, considering the limited sample size, we have employed the exactTest function from the “edgeR” R package to identify differential expressed genes[2, 3]. “edgeR” R package is specifically designed for RNA-seq count data, and its exactTest function could be regarded as a negative binomial version of Fisher’s exact test, making it particularly suitable for two-group comparisons with small sample size[3-5]. We also used stricter thresholds to enhance the statistical power[3, 6] (Page 24 Lines 594-600). Moreover, the final DEG set was obtained by intersecting results with another dataset of larger sample size, which may partly compensate for this limitation. We sincerely apologize for the insufficient clarity in our original explanation. We have clarified the use of exactTest in the Methods section (Page 24 Lines 603-605).

The single cell data were also obtained from the GEO database (GSE103224). Although the number of patient samples was limited, the total cell number was sufficient. We have employed rigorous quality control to minimize technical bias, and have compared gene expression patterns between single cell RNA-seq and bulk RNA-seq to ensure the reliability (Page 13 Figure 6B). Additionally, we used the BayesPrism to infer cell proportion, which incorporates large public cohort (TCGA bulk RNA-seq data), thereby partly compensating for the insufficient number of samples in GSE103224[7, 8]. And this dataset was a widely recognized resource with label annotation, which made our results more objective[9]. We acknowledge that our description of sample size and data limitations may not have been sufficiently clear, and we have revised the Discussion in the revised manuscript (Page 21, Lines 525-527).

Comment 2: The study mostly relies on the bioinformatic predictions, which need a huge dataset and a large sample size. The multiomics data are impressive but rely on in-silico modeling. Most wet lab experiments are needed to strengthen the link between CCL5 and TEX.

Response 2: Thanks for pointing out our shortcomings. We acknowledge that our experimental validation was primarily based on RT-qPCR analysis, without further validation.

Our study primarily relies on multiomics data and bioinformatic analysis, which contributed to our exploratory research. The main purpose of our study was to identify potential key genes and explore their immunological significance through multiomics analysis. We also analysed the immune tumor microenvironment and drug sensitivity of GBM, providing a reference for further research or clinical practice. After obtaining results from the TCGA database, we further performed external validation in Chinese cohort (CGGA) and proteomics data to enhance the reliability and generalizability of our findings[10]. The sample size of these data was sufficient. Although computational analysis could not completely replace wet lab experiments, the integration of multiomics data and statistical verification supported that our conclusions were robust at a preliminary level. After identifying CCL5 as a potential target, we performed in vitro experiments and RT-qPCR analysis to provide further evidence supporting its association with TEX. RT-qPCR had high sensitivity, specificity, and reproducibility, making it the gold standard for gene expression analysis and validating transcriptome datasets[11-14]. Our experiments were designed only to verify the feasibility of this target, because the data collected in this study did not sufficiently support an in-depth investigation of underlying mechanisms, such as cell communication. Despite the limitations in experimental validation, this study identified a series of key TEX-related genes through multiomics analysis and explored their potential associations with prognosis, immune microenvironment, and drug sensitivity. These findings provided important insights for the investigation of the GBM immune microenvironment and the development of novel therapeutic strategies.

We will further investigate the underlying biological mechanisms in the future through more analysis and experiments. And we clearly acknowledge in the Discussion (Page 22 Lines 542-546) that the current experimental validation was limited.

Furthermore, we have refined the language and improved the logical flow to ensure the rigor of our conclusions. 

Comment 3: The paper lacks the mechanistic depth to conclude the CCL5 and TEX via NFkB signaling. The author should perform the experiment related to phosphorylation or NfkB reporter activity to directly correlate in T cells, without CCL5 knockdown. Also, the author can check the activity of the exhaustion markers by treating CCL5 high T cells with NFkB inhibitors. There are various experiments that need to be included in this manuscript to conclude CCL5 as a Prognostic and Therapeutic Target.

Response 3: We sincerely appreciate your suggestion on improving our experiments. We apologize for the lack of in-depth mechanism experiments.

The primary focus of this study was exploratory analysis, aiming to utilize multiomics data to uncover the impact of TEX-related genes on various aspects of GBM, such as potential functions, tumor microenvironment, and drug sensitivity. We fully recognize that mechanism experiments are currently insufficient, and we have emphasized this limitation in the discussion section of the manuscript.

Although direct mechanism experiments were limited, we have provide several indirect evidence supporting the potential link between CCL5 and NFκB pathway: First, pathway enrichment analysis indicated that NFκB signaling pathway was significantly different between high CCL5 expression group and low CCL5 expression group (Page 19 Figure 10G). Second, we referred to the pathway maps from KEGG database, which illustrate the molecular interaction networks. CCL5-NFκB pathway interaction has been supported by KEGG pathway analysis[15-17]. Third, previous studies have reported correlations between CCL5 and NFκB in tumor immune regulation, which are consistent with our computational findings[18, 19]. And some studies have demonstrated NFκB’s crucial role in promoting T cell exhaustion[17, 20]. We have revised and added the above citations and content in the revised manuscript (Page 18 Lines 402-403 and Page 22 Lines 537-538).

Due to limitations in experimental conditions and facilities, we are unable to perform further validation experiments. But we fully agree with your comments and plan to perform more mechanism experiments in future studies to elucidate the relationship between CCL5 and NFκB signaling pathway. This is the limitation of our study, and we have also discussed in the revised manuscript (Page 21 Lines 518-520 and Page 22 Lines 542-548).

Comment 4: The various models' data included in the paper seem to be overfitting, though the risk model used many approaches. The clinical utility of this is questionable, and the author should include the decision curve analysis for better understanding.

Response 4: Thank you very much for your advice. We acknowledge that this part of our study should be improved. As you suggested, we have employed decision curve analysis on four models: two based on TCGA (One survival model includs only the risk group, the other survival model includs both risk stratification and other clinical information) and two based on CGGA (One survival model includs only the risk group, the other survival model includs both risk stratification and other clinical information).

In the TCGA cohort (a–d), the decision curve analysis demonstrated that our two survival models yielded a higher net benefit than both the “treat none” and “treat all” strategies in most cases, especially in the early stages of GBM (at 400 days). However, as time passes and the disease progresses, the strategy that all patients receive intervention yields increasingly benefit. At the end-stage of GBM, our models show clear advantages over the “All”strategy only at higher threshold probability. However, considering the general treatment process for tumors, this phenomenon is reasonable. Additionally, we have found that the survival model incorporating clinical information performed better than survival model that only included risk group across a wider range of risk thresholds. This suggested that survival model includs both risk group and other clinical information was more suitable as a prognostic model. Similarly, in the CGGA cohort (e–h), the model maintained stable net benefit, and conclusions obtained from CGGA were generally consistent with those from TCGA.

Due to the limited length of manuscipt and it’s figures, we have placed the relevant figures in the supplementary materials (Supplementary Materials, on Page 8 Figure S1). We have also added and revised the relevant contents in the main text. The modifications were added on Page 8, Lines 214-222, Page 20 Lines 469-471,and Page 25 Lines 651-653,  in the revised manuscript.

The corresponding results and figures are also dispalyed below.

FIGURE S1. Decision curve analysis on four survival models.

(a) Decision curves of survival models based on TCGA Set at 400 days, 800 days, 1200 days, and 1600 days, respectively. Dark gray represents survival model includs only risk group, orange represents survival model includs both risk stratification and other clinical information, blue represents the reference that all patients receive intervention, red represents the reference that no patients receive intervention.

(b) Decision curves of survival models based on CGGA Set at 400 days, 800 days, 1200 days, and 1600 days, respectively. Dark gray represents survival model includs only risk group, orange represents survival model includs both risk stratification and other clinical information, blue represents the reference that all patients receive intervention, red represents the reference that no patients receive intervention.

Comment 5: Single-cell pictures are very blurred. Please try to get more focused pictures.

Response 5: Thank you for this helpful comment! We apologize for this concern about clarity of our figures. The original figure 6 was very large, so we resized it for submission. However, our scaling was not optimal, resulting in the current file size of only 3.24 MB. We have rescaled the original figure, and the newly uploaded figure is 22.3 MB in TIF format.

The modifications were added on Page 13 Figure 6 in the revised manuscript. The corresponding results is also dispalyed below.

References:

  1. Kirchmair A, Nemati N, Lamberti G, Trefny M, Krogsdam A, Siller A, Hortnagl P, Schumacher P, Sopper S, Sandbichler A, Zippelius A, Ghesquiere B, Trajanoski Z. (13)C tracer analysis reveals the landscape of metabolic checkpoints in human CD8(+) T cell differentiation and exhaustion. Front Immunol, 2023,14, 1267816.
  2. Robinson MD, McCarthy DJ, Smyth GK. edgeR: a Bioconductor package for differential expression analysis of digital gene expression data. Bioinformatics, 2010,26(1), 139-40.
  3. Chen Y, Lun AT, Smyth GK. From reads to genes to pathways: differential expression analysis of RNA-Seq experiments using Rsubread and the edgeR quasi-likelihood pipeline. F1000Res, 2016,5, 1438.
  4. Li C-I, Su P-F, Shyr Y. Sample size calculation based on exact test for assessing differential expression analysis in RNA-seq data. BMC Bioinformatics, 2013,14(1), 357.
  5. Li D, Zand MS, Dye TD, Goniewicz ML, Rahman I, Xie Z. An evaluation of RNA-seq differential analysis methods. PLOS ONE, 2022,17(9), e0264246.
  6. Du B, Zhang F, Zhou Q, Cheng W, Yu Z, Li L, Yang J, Zhang X, Zhou C, Zhang W. Joint analysis of the metabolomics and transcriptomics uncovers the dysregulated network and develops the diagnostic model of high-risk neuroblastoma. Sci Rep, 2023,13(1), 16991.
  7. Chu T, Wang Z, Pe'er D, Danko CG. Cell type and gene expression deconvolution with BayesPrism enables Bayesian integrative analysis across bulk and single-cell RNA sequencing in oncology. Nat Cancer, 2022,3(4), 505-517.
  8. Tran KA, Addala V, Johnston RL, Lovell D, Bradley A, Koufariotis LT, Wood S, Wu SZ, Roden D, Al-Eryani G, Swarbrick A, Williams ED, Pearson JV, Kondrashova O, Waddell N. Performance of tumour microenvironment deconvolution methods in breast cancer using single-cell simulated bulk mixtures. Nature Communications, 2023,14(1), 5758.
  9. Yuan J, Levitin HM, Frattini V, Bush EC, Boyett DM, Samanamud J, Ceccarelli M, Dovas A, Zanazzi G, Canoll P, Bruce JN, Lasorella A, Iavarone A, Sims PA. Single-cell transcriptome analysis of lineage diversity in high-grade glioma. Genome Med, 2018,10(1), 57.
  10. Zhao Z, Zhang KN, Wang Q, Li G, Zeng F, Zhang Y, Wu F, Chai R, Wang Z, Zhang C, Zhang W, Bao Z, Jiang T. Chinese Glioma Genome Atlas (CGGA): A Comprehensive Resource with Functional Genomic Data from Chinese Glioma Patients. Genomics Proteomics Bioinformatics, 2021,19(1), 1-12.
  11. de Brito MWD, de Carvalho SS, Mota MBdS, Mesquita RD. RNA-seq validation: software for selection of reference and variable candidate genes for RT-qPCR. BMC Genomics, 2024,25(1), 697.
  12. Bustin S. Quantification of mRNA using real-time reverse transcription PCR (RT-PCR): trends and problems. Journal of Molecular Endocrinology, 2002,29(1), 23-39.
  13. Liang L, He Z, Yu H, Wang E, Zhang X, Zhang B, Zhang C, Liang Z. Selection and Validation of Reference Genes for Gene Expression Studies in Codonopsis pilosula Based on Transcriptome Sequence Data. Scientific Reports, 2020,10(1), 1362.
  14. Chung Y-S, Lee N-J, Woo SH, Kim J-M, Kim HM, Jo HJ, Park YE, Han M-G. Validation of real-time RT-PCR for detection of SARS-CoV-2 in the early stages of the COVID-19 outbreak in the Republic of Korea. Scientific Reports, 2021,11(1), 14817.
  15. Kanehisa M, Furumichi M, Tanabe M, Sato Y, Morishima K. KEGG: New perspectives on genomes, pathways, diseases and drugs. Nucleic acids research, 2016,45.
  16. Kanehisa M, Furumichi M, Sato Y, Matsuura Y, Ishiguro-Watanabe M. KEGG: biological systems database as a model of the real world. Nucleic Acids Research, 2024,53(D1), D672-D677.
  17. Grinberg-Bleyer Y, Oh H, Desrichard A, Bhatt DM, Caron R, Chan TA, Schmid RM, Klein U, Hayden MS, Ghosh S. NF-kappaB c-Rel Is Crucial for the Regulatory T Cell Immune Checkpoint in Cancer. Cell, 2017,170(6), 1096-1108 e13.
  18. An G, Wu F, Huang S, Feng L, Bai J, Gu S, Zhao X. Effects of CCL5 on the biological behavior of breast cancer and the mechanisms of its interaction with tumor‑associated macrophages. Oncol Rep, 2019,42(6), 2499-2511.
  19. Cui ZY, Park SJ, Jo E, Hwang I-H, Lee K-B, Kim S-W, Kim DJ, Joo JC, Hong SH, Lee M-G, Jang I-S. Cordycepin induces apoptosis of human ovarian cancer cells by inhibiting CCL5-mediated Akt/NF-κB signaling pathway. Cell Death Discovery, 2018,4(1), 62.
  20. Pichler AC, Carrie N, Cuisinier M, Ghazali S, Voisin A, Axisa PP, Tosolini M, Mazzotti C, Golec DP, Maheo S, do Souto L, Ekren R, Blanquart E, Lemaitre L, Feliu V, Joubert MV, Cannons JL, Guillerey C, Avet-Loiseau H, Watts TH, Salomon BL, Joffre O, Grinberg-Bleyer Y, Schwartzberg PL, Lucca LE, Martinet L. TCR-independent CD137 (4-1BB) signaling promotes CD8(+)-exhausted T cell proliferation and terminal differentiation. Immunity, 2023,56(7), 1631-1648 e10.

Reviewer 3 Report

Comments and Suggestions for Authors

The authors made use of multiomics, identifying CCL5 as a prognostic and therapeutic target in glioblastoma. The authors have presented extensive data. 

However, few clarifications are required.

  1. Authors use GWAS on glioma to identify relationship between glioma and increased proportion of exhausted non-naive T cells. Then they directly went for glioblastoma. How/why did they only choose glioblastoma? What about other gliomas?
  2. Figure 5 I & J. The authors have IL-18 and CCL5 protein expression on brain cancer (gliomas) as the highest but the legend mentions GBM. Gliomas can be of many types (astrocytomas, oligodendrogliomas, ependymomas, DMG, etc). 
  3. Figure 6 E and figure 7 D shows highest association of CCL5 with myeloid and macrophages (specifically M2, which is rather interesting, since monocytes [second highest] are known to be producers of CCL5,) respectively. T cells have minimal correlation. Yet for verification purposes they did deletion on T cells to show CCL5 gene deletion results in less exhaustion makers expression, which is an interesting finding. It would be more appropriate to delete CCR5 or its downstream signaling process in the T cells to show correlation with T cells exhaustion. Based on the data provided, logically myeloid cells or monocytes are producing CCL5 which is probably acting on T cells.
  4. Moreover, initially authors have tried to show exhaustion on CD8+ T cells using MR (table 1), but in rest of the figures they show only on total T cells, why? And in figure 10 authors only show total T cells.
  5. In continuation, line 402, authors mention that deletion of CCL5 slows the process of T cell exhaustion. However no data is provided with later time points showing exhaustion markers are equal to controls (data shown fig 10, is RT/qPCR after 15 days incubation with CD3/28 magnetic beads)
  6. Line 283 and 284, 'However, their high expression might lead to T cell exhaustion, with their capacity to induce exhaustion potentially outweighing their ability to recruit T cells'. Do authors have any reference for it or any experimental proof?
  7. Figure 10 D-G, label Y axis.
  8. In figure 10, authors have used in-vitro exhaustion experimental model for proof of concept. It would be more relevant to use in-vivo model (GBM model) for proof of concept that CCL5 deletion results in less exhausted T cells.

Author Response

Dear Reviewer 3,

Thank you very much for your comments regarding our manuscript entitled Multiomics Investigation of Exhausted T cells in Glioblastoma Tumor Microenvironment: CCL5 as a Prognostic and Therapeutic Target [Manuscript ID: ijms-3884261].

Thank you for your letter dated Sep 10. We are truly grateful to your critical comments and thoughtful suggestions on how to improve our manuscript.

The valuable comments and criticisms have provided important guidance for our research and the revision of our manuscript. Based on these comments and suggestions, we have made appropriate modifications on the original manuscript, then we hope to meet your approval. In the revised manuscript, all modifications have been highlighted in red. The reviewer’s comments are reproduced, and our responses are provided directly below each comment in blue. The corresponding revisions are addressed point by point in the main text.

Once again, we deeply appreciate the time and effort dedicated by you. All revisions have been carefully implemented as highlighted in the revised manuscript (marked in red).

Sincerely,

Lei Cao

Overall comments:

The authors made use of multiomics, identifying CCL5 as a prognostic and therapeutic target in glioblastoma. The authors have presented extensive data. 

Response: We would like to thank you for your recognition of our work and helpful comment, which has significantly improved the presentation of our manuscript. Furthermore, your insightful suggestions have provided valuable guidance for our future research, and we sincerely appreciate your comments again.

Comment 1: Authors use GWAS on glioma to identify relationship between glioma and increased proportion of exhausted non-naive T cells. Then they directly went for glioblastoma. How/why did they only choose glioblastoma? What about other gliomas?

Response 1: Thank you for your attention to the process of Mendelian randomization in our study.

We selected glioblastoma (GBM) as the focus of our study for the following three reasons. First, GBM is the most common and aggressive type of glioma, with a median survival time shorter than that of other gliomas[1, 2]. Second, T cell exhaustion generally occurs after disease and persistent antigen stimulation[3-5], which made us particularly interested in the T cell exhaustion and immune microenvironment of GBM, the most aggressive glioma. Third, unlike other gliomas, the first-line chemotherapeutic drug for GBM is limited to TMZ[6, 7], and we were particularly interested in exploring drug sensitivity in GBM to provide a reference for other researches.

T cell exhaustion may also occur in other gliomas, but we have not explored their specific immune microenvironments. We hope to conduct further analysis in the future to extend our conclusions to other gliomas.

Comment 2: Figure 5 I & J. The authors have IL-18 and CCL5 protein expression on brain cancer (gliomas) as the highest but the legend mentions GBM. Gliomas can be of many types (astrocytomas, oligodendrogliomas, ependymomas, DMG, etc). 

Response 2: Thank you for the suggestion regarding our figures and text. We sincerely apologize for the insufficient clarity in our original explanation.

We have confirmed the definition of brain cancer and revised the relevant figures and words. Specifically, We have revised the X-axis labels of Figure 5 I & J, changing “brain cancer” to “GBM” (Page 11 Figure 5). Additionally, we have resized Figure 5 to improve its clarity. The corresponding results is also dispalyed below. We have also revised and polished the text. Specifically, on Page 10 Lines 242-246.

Figure 5. Evaluation and validation of GBM prognostic risk model in TCGA Set and CGGA external validation Set. (A) Kaplan-Meier curve in TCGA Set identifying two risk groups. (B-C) Survival curves for DSS and PFI in TCGA patients with two risk groups. (D) Kaplan-Meier curve in CGGA external validation Set using the same criteria of risk. (E) Time-dependent Receiver Operating Characteristic (ROC) curve in TCGA Set. (F) Time-dependent ROC curve in CGGA external validation Set. (G) Forest plot combined with clinical factors and risk group information in TCGA Set. (H) Forest plot combined with clinical factors and risk group information in CGGA Set. (I-J) Protein expression of IL18 and CCL5 in pan cancer.

Comment 3: Figure 6 E and figure 7 D shows highest association of CCL5 with myeloid and macrophages (specifically M2, which is rather interesting, since monocytes [second highest] are known to be producers of CCL5,) respectively. T cells have minimal correlation. Yet for verification purposes they did deletion on T cells to show CCL5 gene deletion results in less exhaustion makers expression, which is an interesting finding. It would be more appropriate to delete CCR5 or its downstream signaling process in the T cells to show correlation with T cells exhaustion. Based on the data provided, logically myeloid cells or monocytes are producing CCL5 which is probably acting on T cells.

Response 3: Thank you for your helpful suggestion. We apologize for not expressing ourselves clearly. Your comment regarding the association of CCL5 with myeloid cells and macrophages is highly insightful and points out the direction for our future research.

In Figure 6E, we aimed to illustrate the relationship between these genes (CCL5, IL18, CXCR6, FCER1G, TNFSF13B) and proportion of 6 cell types. And intergroup comparisons in Figure 6F showed no significant difference in the proportion of T cells between the high- and low-risk groups. This led us to propose that these five genes might affect the immune microenvironment by impairing T cell function. Subsequently, we employed many immune cell infiltration methods (CIBERSORT, TIMER, EPIC, MCPcounter, xCell) to analyse the proportions of T cell subtypes and other immune cells, finding significant differences in exhausted T cells between the high- and low-risk groups. These results did not indicate which cells produce more CCL5. Our unclear expressing may have caused this misunderstanding, so we have revised the relevant sections on Page 12 Lines 267-274, Page 13 Caption for Figure 6 and Page 25 LInes 665-666.

During the selection of hub genes, we used pure T cell populations (TEX and Teff, Figure 2C), suggesting that CCL5 exhibited differential expression within T cells. Moreover, CCL5 receptors include CCR1, CCR3, CCR4 and CCR5[8], which is supported by KEGG pathway analysis. Some of these axis have been validated in other cancers[9-11]. Based on our previous analysis, we could not determine which specific axis was responsible, and only confirmed that CCL5 was a reliable biomarker. Other previous studies have also discussed CCL5, concluding that normal T cells express and secrete CCL5[12, 13], and its expression may increase under disease conditions[14-16]. Therefore, in theory, our in-vitro exhaustion experiments in were feasible.

Your insightful comments are highly valuable. As this manuscript is our first study on GBM, your comments have clarified future directions for our research. We will focus more on cell communication, collecting new and more accurate data to explore and validate the role of CCL5 in cell communication (such as T cells and NK cells or myeloid). We have also added this point in the revised manuscript (Page 21 Lines 485-487 and Page 22 Lines 546-548).

Comment 4: Moreover, initially authors have tried to show exhaustion on CD8+ T cells using MR (table 1), but in rest of the figures they show only on total T cells, why? And in figure 10 authors only show total T cells.

Response 4: Thank you very much for your comment. In our study, the MR analysis (Table 1) focused on CD8+ T cell exhaustion, which is generally considered the major population in exhausted T cells[4, 17, 18]. Alireza stated in the study that Exhaustion  often restricted to CD8+ T cells responses in the literature, although CD4+ T cells had also been reported to be functionally exhausted in certain chronic infections[19]. Therefore, we used CD8+ T cell exhaustion as a representative measure of overall T cell exhaustion. Therefore, we utilized several datasets to to improve the reliability.

For the subsequent experiments (Figure 10), due to the lack of facilities to perform flow cytometry, we only measured total T cells and were unable to discuss subtypes separately. Despite this limitation, we believe the overall conclusions regarding T cell exhaustion remain valid. We will consider the two subtypes of exhausted T cells in future studies and investigate their cell communication, while also explore new experimental techniques.

We have added these descriptions and limitations on Page 20 Lines 457-459, Page 22 Lines 542-544 and Page 23 Lines 569-570 in the revised manuscript.

Comment 5: In continuation, line 402, authors mention that deletion of CCL5 slows the process of T cell exhaustion. However no data is provided with later time points showing exhaustion markers are equal to controls (data shown fig 10, is RT/qPCR after 15 days incubation with CD3/28 magnetic beads).

Response 5: Thank you for your interest in the RT-qPCR experiment. The in-vitro exhaustion experimental model refered to previous studies. Mirko Corselli’ study developed two in-vitro models mimicking either chronic or transient T-cell stimulation. Chronic stimulation was achieved by continuously stimulating T cells with recombinant human IL-2 (rhIL-2) and CD3/CD28 beads for 14 days. Transient stimulation was achieved by stimulating T cells with rh-IL2 and CD3/CD28 beads for 3 days, followed by resting in the presence of rhIL-2 for additional 11 days[20]. Manzhi Zhao’s study developed a novel  in-vitro exhaustion model, and T cells were stimulated for only 5 days with 10ng/ml OVA peptide in the presence of IL-15 and IL-7[21]. Jennifer’ study found that inhibitory receptors expression of their in-vitro exhaustion model at day 7 and day 10 was comparable, suggesting little increase in exhaustion[22]. Considering the establishment of in-vitro exhaustion models in the above studies, we set the experiment duration to 15 days. In the Discussion, we also acknowledged the limitations of our experiment. Your suggestion has further reminded us to pay closer attention to the setting of experimental duration in future work.

We have also inserted the above references on Page 26 Lines 713-716 in the revised manuscript.

Comment 6: Line 283 and 284, 'However, their high expression might lead to T cell exhaustion, with their capacity to induce exhaustion potentially outweighing their ability to recruit T cells'. Do authors have any reference for it or any experimental proof?

Response 6: Thank you for this helpful comment. We apologize for the imprecise statements and the lack of supporting references in manuscript.

We had already read relevant literature but did not cite them. The references are as follows:

Some studies have found that high expression of CCL5 is associated with T cell exhaustion in other cancers[23, 24]. On the other hand, some studies have shown that CCL5 could recruits T cells and other immune cells into the tumor microenvironment and reshapes the TME[25, 26]. Moreover, team of Zhang explicitly discovered that CCL5 plays dual roles in colorectal cancer progression[25].

IL18 is a cytokine that is involved in the activation of immune responses[27]. Meanwhile, there is a study has found that IL18 signaling pathway is associated with T cell exhaustion[28].

CXCR6 and FCER1G have been found to be involved in the trafficking and activation of immune cells[29, 30], while TNFSF13B has been implicated in the regulation of immune responses[31].

We have revised this part on Page 12 Lines 296-299, Page 20 Lines 462-468 and Pages 20-21 Line 483-488 in the revised manuscript. And we have added a detailed description and citations about these gense on 20 Lines 462-468 in the revised manuscript.

Comment 7: Figure 10 D-G, label Y axis.

Response 7: Thank you for this helpful comment. As you suggested, we have verified the variable labels corresponding to the Y axis in these four figures and have made modified the relevant content in these figures. In addition, we have checked other figures for similar questions and have made necessary corrections.

We have added variable labels of Y axis on Page 19 Figure 10 D-G, Page 15 Figure 8 A-H. The new figures are also displayed below.

Comment 8: In figure 10, authors have used in-vitro exhaustion experimental model for proof of concept. It would be more relevant to use in-vivo model (GBM model) for proof of concept that CCL5 deletion results in less exhausted T cells.

Response 8: Thanks for pointing out our shortcomings. We acknowledge that our experimental validation was primarily based on RT-qPCR analysis, without further validation.

Previous studies suggest that, under restricted experimental conditions and facilities, in-vitro models are more practical and feasible, as exhausted T cells are difficult to observe in-vivo model. The references are as follows:

In Jennifer’study, it was explicitly pointed that, despite their utility, in-vivo models have limitations in efficiency and cellular yield, despite their utility[22]. This study also noted that mouse models of T cell exhaustion often generate small numbers of Tex[22]. A similar statement was also found in Zhao’s paper. The paper pointed out that inducing CTL exhaustion in-vivo is time-consuming, requiring more than 30 days and yields limited numbers of exhausted cells as experimental material. Most importantly the in-vivo milieu in these models is characterized by inflammation, high viral loads, suppressive cytokines or cells; all of which can obscure the phenotype of exhausted CTL and therefore make it hard to dissect true exhaustion-associated changes[21]. The above studies suggested that in-vitro models were more practical, however, conducting in-vitro exhaustion experiments also has certain advantages and could provide sufficient evidence. Additionally, our team currently lacks the facilities required for in-vivo experiments, so we used exhaustion experimental model to validate our concept. In the Discussion (Page 21 Lines 518-521 and Page 22 Lines 543-549), we also acknowledged the limitations of our experiment and indicated that additional experiments (such as animal models) will be explored in the future.

References:

  1. Alexander BM, Cloughesy TF. Adult Glioblastoma. J Clin Oncol, 2017,35(21), 2402-2409.
  2. Woroniecka K, Chongsathidkiet P, Rhodin K, Kemeny H, Dechant C, Farber SH, Elsamadicy AA, Cui X, Koyama S, Jackson C, Hansen LJ, Johanns TM, Sanchez-Perez L, Chandramohan V, Yu YA, Bigner DD, Giles A, Healy P, Dranoff G, Weinhold KJ, Dunn GP, Fecci PE. T-Cell Exhaustion Signatures Vary with Tumor Type and Are Severe in Glioblastoma. Clin Cancer Res, 2018,24(17), 4175-4186.
  3. Blank CU, Haining WN, Held W, Hogan PG, Kallies A, Lugli E, Lynn RC, Philip M, Rao A, Restifo NP, Schietinger A, Schumacher TN, Schwartzberg PL, Sharpe AH, Speiser DE, Wherry EJ, Youngblood BA, Zehn D. Defining 'T cell exhaustion'. Nat Rev Immunol, 2019,19(11), 665-674.
  4. Wherry EJ. T cell exhaustion. Nat Immunol, 2011,12(6), 492-9.
  5. Chu T, Wu M, Hoellbacher B, de Almeida GP, Wurmser C, Berner J, Donhauser LV, Ann-Katrin G, Lin S, Cepeda-Mayorga JD, Kilb, II, Bongers L, Toppeta F, Strobl P, Youngblood B, Schulz AM, Zippelius A, Knolle PA, Heinig M, Hackstein CP, Zehn D. Precursors of exhausted T cells are preemptively formed in acute infection. Nature, 2025.
  6. Horbinski C, Nabors LB, Portnow J, Baehring J, Bhatia A, Bloch O, Brem S, Butowski N, Cannon DM, Chao S, Chheda MG, Fabiano AJ, Forsyth P, Gigilio P, Hattangadi-Gluth J, Holdhoff M, Junck L, Kaley T, Merrell R, Mrugala MM, Nagpal S, Nedzi LA, Nevel K, Nghiemphu PL, Parney I, Patel TR, Peters K, Puduvalli VK, Rockhill J, Rusthoven C, Shonka N, Swinnen LJ, Weiss S, Wen PY, Willmarth NE, Bergman MA, Darlow S. NCCN Guidelines(R) Insights: Central Nervous System Cancers, Version 2.2022. J Natl Compr Canc Netw, 2023,21(1), 12-20.
  7. Hart MG, Garside R, Rogers G, Stein K, Grant R. Temozolomide for high grade glioma. Cochrane Database Syst Rev, 2013,2013(4), CD007415.
  8. Appay V, Rowland-Jones SL. RANTES: a versatile and controversial chemokine. Trends in Immunology, 2001,22(2), 83-87.
  9. Sutton A, Friand V, Papy-Garcia D, Dagouassat M, Martin L, Vassy R, Haddad O, Sainte-Catherine O, Kraemer M, Saffar L, Perret GY, Courty J, Gattegno L, Charnaux N. Glycosaminoglycans and their synthetic mimetics inhibit RANTES-induced migration and invasion of human hepatoma cells. Mol Cancer Ther, 2007,6(11), 2948-58.
  10. Choi SW, Hildebrandt GC, Olkiewicz KM, Hanauer DA, Chaudhary MN, Silva IA, Rogers CE, Deurloo DT, Fisher JM, Liu C, Adams D, Chensue SW, Cooke KR. CCR1/CCL5 (RANTES) receptor-ligand interactions modulate allogeneic T-cell responses and graft-versus-host disease following stem-cell transplantation. Blood, 2007,110(9), 3447-55.
  11. Yang H, Cai J, Huang X, Zhan C, Lu C, Gu J, Ma T, Zhang H, Cheng T, Xu F, Ge D. Gram-Negative Microflora Dysbiosis Facilitates Tumor Progression and Immune Evasion by Activating the CCL3/CCL5–CCR1–MAPK–PD-L1 Pathway in Esophageal Squamous Cell Carcinoma. Molecular Cancer Research, 2025,23(1), 71-85.
  12. Suffee N, Richard B, Hlawaty H, Oudar O, Charnaux N, Sutton A. Angiogenic properties of the chemokine RANTES/CCL5. Biochemical Society Transactions, 2011,39(6), 1649-1653.
  13. Ortiz BD, Krensky AM, Nelson PJ. Kinetics of transcription factors regulating the RANTES chemokine gene reveal a developmental switch in nuclear events during T-lymphocyte maturation. Mol Cell Biol, 1996,16(1), 202-10.
  14. Mori N, Krensky AM, Ohshima K, Tomita M, Matsuda T, Ohta T, Yamada Y, Tomonaga M, Ikeda S, Yamamoto N. Elevated expression of CCL5/RANTES in adult T-cell leukemia cells: possible transactivation of the CCL5 gene by human T-cell leukemia virus type I tax. Int J Cancer, 2004,111(4), 548-57.
  15. Dong G, Fan F, He Y, Luo Y, Yu J, Liang P. T-Lymphocyte Gene-Regulated CCL5 and Its Association with Extrahepatic Metastasis in Hepatocellular Carcinoma. J Hepatocell Carcinoma, 2023,10, 1267-1279.
  16. van Elsas MJ, Middelburg J, Labrie C, Roelands J, Schaap G, Sluijter M, Tonea R, Ovcinnikovs V, Lloyd K, Schuurman J, Riesenfeld SJ, Gajewski TF, de Miranda NFCC, van Hall T, van der Burg SH. Immunotherapy-activated T cells recruit and skew late-stage activated M1-like macrophages that are critical for therapeutic efficacy. Cancer Cell, 2024,42(6), 1032-1050.e10.
  17. Wherry EJ, Ha S-J, Kaech SM, Haining WN, Sarkar S, Kalia V, Subramaniam S, Blattman JN, Barber DL, Ahmed R. Molecular Signature of CD8+ T Cell Exhaustion during Chronic Viral Infection. Immunity, 2007,27(4), 670-684.
  18. Song Q, Hou Y, Zhang Y, Liu J, Wang Y, Fu J, Zhang C, Cao M, Cui Y, Zhang X, Wang X, Zhang J, Liu C, Zhang Y, Wang P. Integrated multi-omics approach revealed cellular senescence landscape. Nucleic Acids Res, 2022,50(19), 10947-10963.
  19. Saeidi A, Zandi K, Cheok YY, Saeidi H, Wong WF, Lee CYQ, Cheong HC, Yong YK, Larsson M, Shankar EM. T-Cell Exhaustion in Chronic Infections: Reversing the State of Exhaustion and Reinvigorating Optimal Protective Immune Responses. Front Immunol, 2018,9, 2569.
  20. Corselli M, Saksena S, Nakamoto M, Lomas III WE, Taylor I, Chattopadhyay PK. Single cell multiomic analysis of T cell exhaustion in vitro. Cytometry Part A, 2022,101(1), 27-44.
  21. Zhao M, Kiernan CH, Stairiker CJ, Hope JL, Leon LG, van Meurs M, Brouwers-Haspels I, Boers R, Boers J, Gribnau J, van IWFJ, Bindels EM, Hoogenboezem RM, Erkeland SJ, Mueller YM, Katsikis PD. Rapid in vitro generation of bona fide exhausted CD8+ T cells is accompanied by Tcf7 promotor methylation. PLoS Pathog, 2020,16(6), e1008555.
  22. Wu JE, Manne S, Ngiow SF, Baxter AE, Huang H, Freilich E, Clark ML, Lee JH, Chen Z, Khan O, Staupe RP, Huang YJ, Shi J, Giles JR, Wherry EJ. In vitro modeling of CD8+ T cell exhaustion enables CRISPR screening to reveal a role for BHLHE40. Science Immunology, 2023,8(86), eade3369.
  23. Zhu Q, Yang Y, Zeng Y, Chen K, Zhang Q, Wang L, Huang Y, Jian S. The significance of CD8(+) tumor-infiltrating lymphocytes exhaustion heterogeneity and its underlying mechanism in diffuse large B-cell lymphoma. Int Immunopharmacol, 2024,137, 112447.
  24. Melese ES, Franks E, Cederberg RA, Harbourne BT, Shi R, Wadsworth BJ, Collier JL, Halvorsen EC, Johnson F, Luu J, Oh MH, Lam V, Krystal G, Hoover SB, Raffeld M, Simpson RM, Unni AM, Lam WL, Lam S, Abraham N, Bennewith KL, Lockwood WW. CCL5 production in lung cancer cells leads to an altered immune microenvironment and promotes tumor development. Oncoimmunology, 2022,11(1), 2010905.
  25. Zhang XF, Zhang XL, Wang YJ, Fang Y, Li ML, Liu XY, Luo HY, Tian Y. The regulatory network of the chemokine CCL5 in colorectal cancer. Ann Med, 2023,55(1), 2205168.
  26. Araujo JM, Gomez AC, Aguilar A, Salgado R, Balko JM, Bravo L, Doimi F, Bretel D, Morante Z, Flores C, Gomez HL, Pinto JA. Effect of CCL5 expression in the recruitment of immune cells in triple negative breast cancer. Scientific Reports, 2018,8(1), 4899.
  27. Kaplanski G. Interleukin-18: Biological properties and role in disease pathogenesis. Immunol Rev, 2018,281(1), 138-153.
  28. Lutz V, Hellmund VM, Picard FSR, Raifer H, Ruckenbrod T, Klein M, Bopp T, Savai R, Duewell P, Keber CU, Weigert A, Chung H-R, Buchholz M, Menke A, Gress TM, Huber M, Bauer C. IL18 Receptor Signaling Regulates Tumor-Reactive CD8+ T-cell Exhaustion via Activation of the IL2/STAT5/mTOR Pathway in a Pancreatic Cancer Model. Cancer Immunology Research, 2023,11(4), 421-434.
  29. Lesch S, Blumenberg V, Stoiber S, Gottschlich A, Ogonek J, Cadilha BL, Dantes Z, Rataj F, Dorman K, Lutz J, Karches CH, Heise C, Kurzay M, Larimer BM, Grassmann S, Rapp M, Nottebrock A, Kruger S, Tokarew N, Metzger P, Hoerth C, Benmebarek MR, Dhoqina D, Grunmeier R, Seifert M, Oener A, Umut O, Joaquina S, Vimeux L, Tran T, Hank T, Baba T, Huynh D, Megens RTA, Janssen KP, Jastroch M, Lamp D, Ruehland S, Di Pilato M, Pruessmann JN, Thomas M, Marr C, Ormanns S, Reischer A, Hristov M, Tartour E, Donnadieu E, Rothenfusser S, Duewell P, Konig LM, Schnurr M, Subklewe M, Liss AS, Halama N, Reichert M, Mempel TR, Endres S, Kobold S. T cells armed with C-X-C chemokine receptor type 6 enhance adoptive cell therapy for pancreatic tumours. Nat Biomed Eng, 2021,5(11), 1246-1260.
  30. Yang R, Chen Z, Liang L, Ao S, Zhang J, Chang Z, Wang Z, Zhou Y, Duan X, Deng T. Fc Fragment of IgE Receptor Ig (FCER1G) acts as a key gene involved in cancer immune infiltration and tumour microenvironment. Immunology, 2023,168(2), 302-319.
  31. Ma T, Meng L, Wang X, Tian Z, Wang J, Liu X, Zhang W, Zhang Y. TNFSF13B and PPARGC1A expression is associated with tumor-infiltrating immune cell abundance and prognosis in clear cell renal cell carcinoma. Am J Transl Res, 2021,13(10), 11048-11064.

Round 2

Reviewer 2 Report

Comments and Suggestions for Authors

All questions were answered. 

Author Response

Dear Reviewer 2,

Thank you very much for your comments regarding our manuscript entitled Multiomics Investigation of Exhausted T cells in Glioblastoma Tumor Microenvironment: CCL5 as a Prognostic and Therapeutic Target [Manuscript ID: ijms-3884261].

We sincerely appreciate your time and effort in reviewing our manuscript. We are very grateful for your valuable comments and suggestions during the review process, which helped us improve the quality and clarity of our work.

We are pleased to know that our responses have met your expectations. Thank you again for your constructive feedback and support.

Sincerely,

Lei Cao

Reviewer 3 Report

Comments and Suggestions for Authors

Authors have done well in addressing comments.

It would interesting to see expression CCL5 receptor  on myeloid cells.

Authors are requested to use uniform annotation for figures. Some places authors have used (Figure 6E) , (Figure 10A,C) while in other place (Figure 8 E,F).

Author Response

Dear Reviewer 3,

Thank you very much for your comments regarding our manuscript entitled Multiomics Investigation of Exhausted T cells in Glioblastoma Tumor Microenvironment: CCL5 as a Prognostic and Therapeutic Target [Manuscript ID: ijms-3884261].

Thank you for your letter dated Oct 06. We are truly grateful to your critical comments and thoughtful suggestions on how to improve our manuscript.

Your suggestion regarding the annotation for figures has been very helpful in improving the overall quality of our manuscript. Based on this suggestion, we have made appropriate modifications on the original manuscript, then we hope to meet your approval. In the revised manuscript, all modifications have been highlighted in red.

Once again, we deeply appreciate the time and effort dedicated by you. All revisions have been carefully implemented as highlighted in the revised manuscript (marked in red).

Sincerely,

Lei Cao

Comment 1: Authors have done well in addressing comments. It would interesting to see expression CCL5 receptor on myeloid cells.

Response 1: We would like to thank you for your recognition of our work and helpful comment, which has significantly improved the presentation of our manuscript. In our future studies, we will focus on the expression and role of CCL5 in other cells, such as myeloid cells and NK cells, in order to further explore the related cell communication mechanisms.

Your insightful suggestions have provided valuable guidance for our future research, and we sincerely appreciate your comments again.

Comment 2: Authors are requested to use uniform annotation for figures. Some places authors have used (Figure 6E) , (Figure 10A,C) while in other place (Figure 8 E,F).

Response 2: Thank you for your helpful comments! We have carefully revised the manuscript to ensure uniform annotation for figures. The modifications were highlighted on Page 2 Lines 72-81, Page 12 Lines 279 and 291 in the revised manuscript.
